# Peptide nano-blanket impedes fibroblasts activation and subsequent formation of pre-metastatic niche

Yi Zhou [1], Peng Ke[1,2], Xiaoyan Bao[1], Honghui Wu[1], Yiyi Xia[1], Zhentao Zhang[1], Haiqing Zhong[1], Qi Dai[1,3], Linjie Wu[1], Tiantian Wang[1], Mengting Lin[1], Yaosheng Li[1], Xinchi Jiang[1], Qiyao Yang[1,3], Yiying Lu[1], Xincheng Zhong[1], Min Han [1,4,5 ✉] & Jianqing Gao[1,4,5 ✉]

There is evidence to suggest that the primary tumor induces the formation of a pre-metastatic niche in distal organs by stimulating the production of pro-metastatic factors. Given the fundamental role of the pre-metastatic niche in the development of metastases, interruption of its formation would be a promising strategy to take early action against tumor metastasis. Here we report an enzyme-activated assembled peptide FR17 that can serve as a "flame-retarding blanket" in the pre-metastatic niche specifically to extinguish the "fire" of tumor-supportive microenvironment adaption. We show that the in-situ assembled peptide nano-blanket inhibits fibroblasts activation, suppressing the remodeling of the metastasis-supportive host stromal tissue, and reversing vascular destabilization and angiogenesis. Furthermore, we demonstrate that the nano-blanket prevents the recruitment of myeloid cells to the pre-metastatic niche, regulating the immune-suppressive microenvironment. We show that FR17 administration effectively inhibits the formation of the pulmonary pre-metastatic niche and postoperative metastasis, offering a therapeutic strategy against pre-metastatic niche formation.

[1] Institute of Pharmaceutics, Zhejiang Province Key Laboratory of Anti-Cancer Drug Research, College of Pharmaceutical Sciences, Zhejiang University, Hangzhou 310058 Zhejiang, PR China. [2] Department of Anesthesiology, Shengli Clinical Medical College, Fujian Medical University, Fuzhou 350001 Fujian, PR China. [3] Department of Radiation Oncology, Key Laboratory of Cancer Prevention and Intervention, The Second Affiliated Hospital, College of Medicine, Zhejiang University, Hangzhou 310058 Zhejiang, PR China. [4] Cancer Center of Zhejiang University, Zhejiang University, Hangzhou 310058 Zhejiang, PR China. [5] Hangzhou Institute of Innovative Medicine, Zhejiang University, Hangzhou 310058 Zhejiang, PR China. ✉email: hanmin@zju.edu.cn; gaojianqing@zju.edu.cn

Though therapeutic outcomes and survival rate for patients with various cancers have been greatly improved in last decades, effective treatments for patients with metastatic cancer are still limited[1]. Though emerging technologies provided early detection of malignant transformation or even warning of high metastatic risk by biomarker screening before the actual occurrence of metastasis in clinic[2–6], clinical treatment on metastasis prevention lags far behind. The contemporary therapeutic strategies against metastasis in clinic, including systemic chemotherapy, radiotherapy and immunotherapy, mainly focus on the later time period of metastasis development, or at least after the arrival and colonization of disseminated tumor cells to the distal organs, which have gained unsatisfied clinical outcomes[7,8]. Observations of the adjuvant chemotherapy resistance[6] and treatment-related toxicities[9] revealed the short-coming of the currently available strategies[10].

Growing evidence illustrated that an inflammatory, neovascularized, immunosuppressive, tumor-supportive microenvironment has emerged before the arrival and colonization of disseminated tumor cells, which is termed as pre-metastatic niche (PMN)[11]. Relevant studies revealed that complex interactions between multiple participators and alteration in regulative pathways energized the construction of PMN, such as primary tumor-derived cytokines and exosomes, myeloid-derived suppressor cells (MDSCs), and the tumor re-educated stromal environment including pre-metastasis associated fibroblasts, destabilized vasculature and extracellular matrix (ECM)[12,13] (Fig. 1a). In 2005, Lyden et al. brought to light the recruitment of VEGFR+ myeloid progenitor cells to PMN by localized fibronectin (FN) deposition[14], which would be over-produced by activated resident fibroblasts. This specific cell population and its subtypes were then unified and classified as MDSCs with its potent capability to suppress immune responses[15], who make major contributions in developing immunosuppressive microenvironment via activation of nitric oxide (NO) signaling or reactive oxygen species (ROS) pathway[16]. What's more, recent studies revealed the irritation of stromal cells especially fibroblasts in primary tumor and in distal site induced by tumor-derived secreted factors via STAT3 signaling[17] and JNK signaling[18] pathways. It is reported that tumor-educated fibroblasts serve to reconstruct ECM, induce angiogenic and pro-inflammatory response of endothelial cells, preparing a tumor-supportive host stroma[19], which gives a clue to the tipping point of PMN initializing as fibroblast activation.

Since PMN was considered as the foundation laid for circulating tumor cells colonization and one of the vital premises to develop metastasis in distant organs, we wonder if the early process of tumor metastasis can be terminated or even totally prevented by interrupting the formation of PMN. To view the entire process of PMN establishment and metastasis development as a progress of the occurrence and the spread of a huge forest fire, it would be more efficient to contain and beat out the local flame by preventing the formation of PMN.

Inspired by the self-assembled peptides found in many natural life processes, researchers have modified, designed and synthesized diverse self-assembled peptides applied as functional biomaterials for various applications[20–22]. In the application on anti-tumor and anti-metastasis therapy, enzyme-responsive self-assembly or ligand-receptor interactions-triggered morphology transforming of the peptide nanofibrils and hydrogel have been developed to induce cell death[23–25], to restrict tumor cell invasion[26], to serve as biocompatible drug delivery platforms[27], to achieve specific targeting of tumor[28] and imaging[29] as well. One of the research branches, which has been widely applied, was based on the β-sheet regions of Amyloid-β (Aβ)[30]. As the original fragment of Aβ$_{16-20}$, KLVFF has been applied as therapeutic agents[26,31], imaging agents[32] or delivery platform[33] in the field of

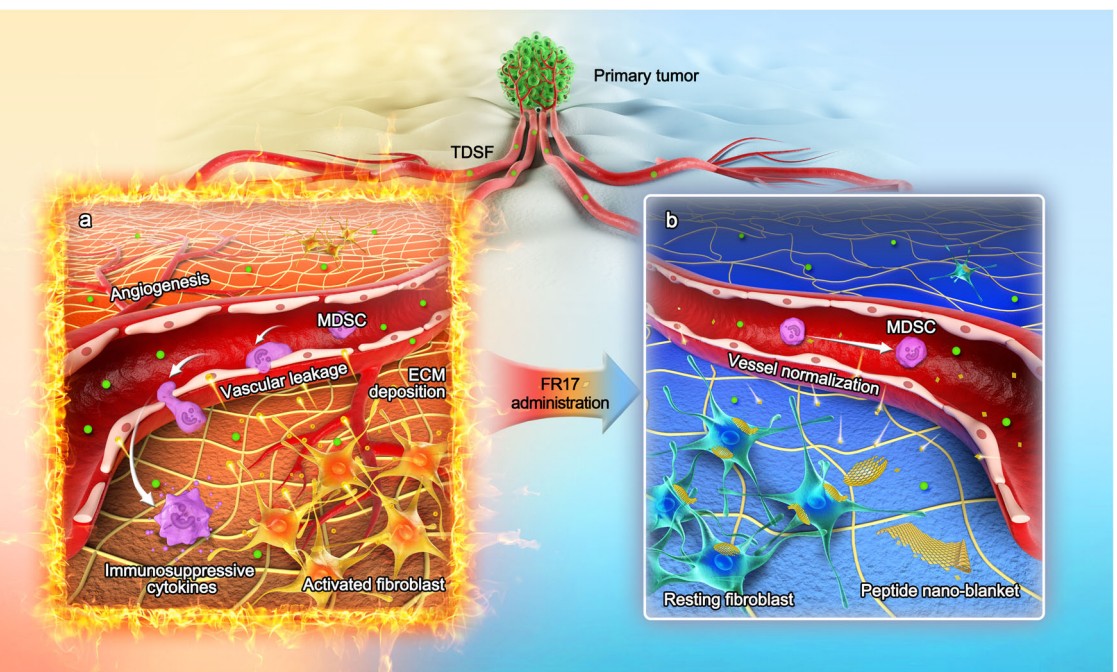

**Fig. 1 Schematic illustration of peptide nano-blanket impedes fibroblasts activation and subsequent formation of pre-metastatic niche. a** Illustration of the pathological process of pre-metastatic niche (PMN) formation. The primary tumor produces pro-metastatic factors, such as tumor-derived secreted factors (TDSF), to induce fibroblast activation in metastatic destination organs. The tumor-educated activated fibroblasts serve to construct a metastasis-supportive host stroma, including extracellular matrix (ECM) deposition and reconstruction, angiogenesis and vascular leakage, as well as myeloid-derived suppressor cells (MDSC) recruitment. **b** After FR17 subcutaneous administration, the in-situ assembled peptide nano-blanket in PMN stromal microenvironment impedes fibroblasts activation so as to retard stromal and vessel pro-metastatic reconstruction, inhibiting MDSC recruitment and metastatic cascades.

Alzheimer's disease[34] and cancer[35]. Further researches cut the peptide fragment down to dipeptide FF to obtain discrete nano-tubes through self-assembly in aqueous solution[36]. Afterwards, felicitous modification and re-designment on FF with perfect biocompatibility has been reported with a wide range of applications[37]. For example, FFKF was developed to construct drug delivery system which can be completely degraded by cathepsin proteases[27]. Yang and his team employed FFFK to develop molecular hydrogel for codelivery of anti-cancer drugs[38]. In another work, the application of FFYK was explored in organelles targeting and cancer cell killing[39]. Some introduced naphthyl group on N-terminal of Phe to favor the intermolecular hydrophobic interactions of FFKY[40].

Here we report an enzyme-activatable assembled peptide FR17 that can serve as a "flame-retarding blanket" at PMN site specifically, containing and suppressing the "fire blaze" of PMN formation to further develop into overt metastasis (Fig. 1b). As a substrate of matrix metalloproteinase 2 (MMP2), FR17 is designed to release self-assembled monomer FG8 to construct peptide nano-blanket in PMN stromal microenvironment. The self-assembly feature of FG8 is provided by its backbone, FFKY, taking advantages of both the self-assembly feature of diphenylalanine structural motif FF with the assistance of Y and the editable site provided by K to combine with hydrophilic fragment via enzyme-cleavable linker. The in-situ assembled peptide nano-blanket impedes fibroblasts activation so as to prevent metastatic cascades. In this work, we demonstrate the subsequent impact of inhibition on fibroblasts activation induced by FR17 intervene, revealing the underlying mechanism on cellular interactions among fibroblasts, vascular endothelial cells and extracellular components, and the intervention on PMN recruited MDSCs via in vitro and in vivo experiments.

## Results

**Peptide nano-blanket transformed from FR17.** The MMP2-activatable self-assembled branched peptide (FR17, FFK(GPLG LAGG-YVDKR)Y) consists of (1) the backbone of a self-assembled peptide domain Phe-Phe-Lys-Tyr (FFKY), a variant of Phe-Phe (FF), which is derived from Aβ[36]; (2) thymopentin (TP5, Arg-Lys-Asp-Val-Tyr, RKDVY), a pentapeptide with perfect hydrophilic property and immune modulation effect, which is applied as the adjunctive therapeutic agent on cancer treatment in clinic to prevent postoperative infection and to activate immune response[41,42]. TP5 was conjugated to the side-chain of Lys (K) in the main-chain FFKY with the MMP2-cleavable peptide linker (Gly-Pro-Leu-Gly-Leu-Ala-Gly-Gly, GPLGLAGG)[43], increasing the hydrophilic property of the entire peptide molecule. Furthermore, the sequence of TP5 in peptide FR17 was replaced by the scrambled pentapeptide of TP5 without immunomodulatory bioactivity, i.e. DVYKR, to form the scrambled group (sFD17, FFK(GPLGLAGG-RKYVD)Y) as peptide assembly control.

Peptides were synthesized via Fmoc solid-phase peptide synthesis technology and the peptide sequences were verified by mass spectra (Supplementary Figs. 1 and 2). When specifically cleaved by MMP2, both FR17 and sFD17 are able to release self-assembled monomer FG8 (FFK(GPLG)Y) (Supplementary Fig. 3), which would spontaneously fold into membrane-like structure, i.e. the peptide nano-blanket, both in neutral and weak acidic conditions, which simulates the pH condition of inflammatory microenvironment in PMN[44,45] (Supplementary Fig. 4). The transmission electron microscopy (TEM) images (Fig. 2a) and the Cryo-TEM image (Supplementary Fig. 5a) showed the lamellar structure of the peptide self-assemblies formed by enzymatic degradation (Supplementary Fig. 5b) dispersed with an average

diameter of ~500 nm (Fig. 2b). The suspected appearance of peptide nano-blanket has been observed in the intercellular substance in PMN lung in vivo (Supplementary Fig. 5c), and the morphology and the size of which ranges from hundreds to thousands nanometers depending on the intercellular space specifically at PMN sites. Besides, the aggregation-induced emission (AIE) effect was employed to monitor the spontaneous aggregation of FG8 in aqueous system (Supplementary Fig. 6). Fluorescent dots observed in the organs' sections revealed the responsive-assembly of the peptide nano-blanket in PMN lung in vivo (Supplementary Fig. 7) with few aggregations in PMN liver and no observation of AIE effect in other organs (Supplementary Fig. 8). Taken together, these results indicated the self-assembly property of FG8 monomer, and MMP2-cleaved release of FG8 from FR17 or sFD17 to form the peptide nano-blanket both in vitro and in vivo.

The peptide assembly relies on the non-bonded interactions, typically hydrogen bonds and π-π stacking, between adjacent peptide molecules. The self-assembling pattern of FG8 changed due to the branch modification with GPLG side-chain. The all-atom molecular dynamics (MD) simulation of FFKY-assemblies and FG8-assemblies gives a microscopic account of the impacts of branch modification on peptide interactions. The MD simulation revealed that there are more intermolecular hydrogen bonds involved in FG8 cluster, which are more ordered and mostly formed between Phe of adjacent FG8 molecules (Supplementary Fig. 9). While the intermolecular hydrogen bonds involved in FFKY-assemblies are less-formed and disordered (Fig. 2c–h).

**The pathological process in the MCM-induced PMN model in vivo.** In order to study the effect of FR17 administration on the process of PMN development, an in vivo PMN model induced by melanoma-conditioned media (MCM) has been established according to the previous research on PMN[14]. Briefly, tumor-derived factors secreted from primary tumor were replaced by MCM intraperitoneal injection to the mice for 10 consecutive days from day 1 to 10. On day 7, when the tumor-supportive microenvironment was successfully established in the lung, B16F10 melanoma cells were intravenously administrated as the simulation of circulative tumor cells (CTC) wandering through blood vessels and some would successfully colonize in the prepared "fire scene" in lung (Supplementary Fig. 10b).

The pathological process in the lung was assessed from various aspects during MCM-induced PMN establishment from day 3 to day 13 (Supplementary Figs. 10 and 11). Important cell derived molecular components and cytokines were closely monitored during the process. The higher expression of the ECM component FN was generated by MCM inducement. Moreover, matrix metalloproteinase 9 (MMP9) and vascular endothelial growth factor a (VEGFa) were up-regulated along with the pathological progress of lung PMN from day 0 to day 10 and reached a plateau on day 10 (Supplementary Fig. 10a). An accordant trend was found on MMP2 level as well (Supplementary Fig. 10d). Meanwhile, the immune cell population analysis during the pathological process in the lung of MCM-induced PMN model was conducted by flow cytometry, which drew our attention to the crime culprit-cell inducing PMN, MDSC, for the recruitment of MDSC increased through the timeline (Supplementary Figs. 11 and 12). It is reported that MDSC contributes a lot in developing the immunosuppressive microenvironment in PMN, preparing more suitable and fertile land for tumor cells to take root in refs. [13,46]. Expression level of the major inflammatory mediator TGF-β1, possibly produced by MDSCs, increased gradually but sharply. Typical biomarkers produced by MDSC to exert immunosuppressive effect, in other words, the

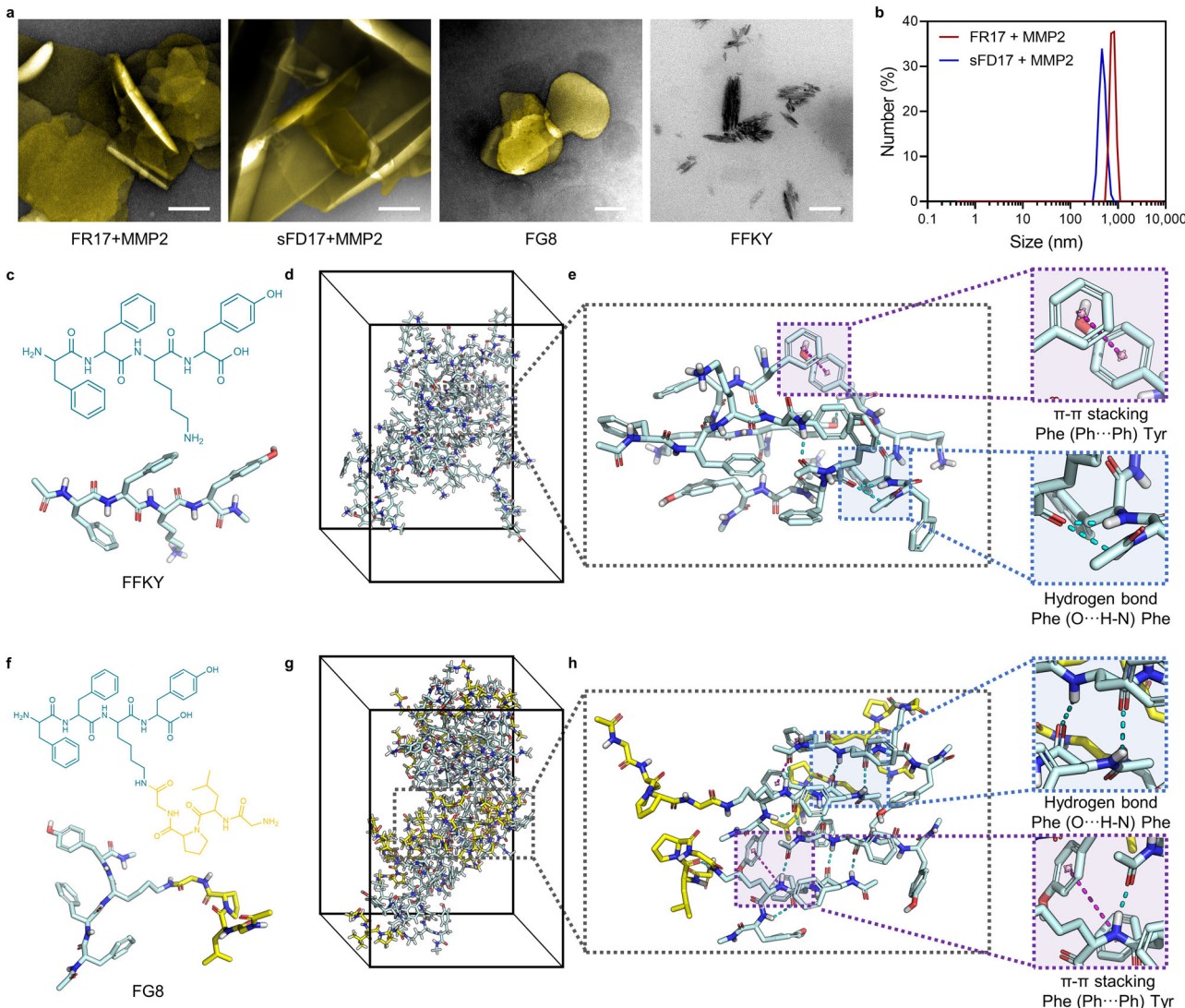

**Fig. 2 Enzyme-activated self-assembly of FR17 and all-atom molecular dynamics (MD) simulation of the self-assembly of FG8. a** TEM images of FR17 or sFD17 (500 μM) treated with MMP2 (scale bar = 200 nm), FG8 (500 μM, scale bar = 100 nm) and FFKY (500 μM, scale bar = 100 nm). The lamellar structure of the peptide nano-blanket was pseudo-colored in gold. **b** Size distribution of FR17 and sFD17 (500 μM) after MMP2 cleavage. **c** Molecular structure of FFKY. **d** The FFKY assemblies generated at $t = 200$ ns of MD simulation of 16 FFKY molecules in water (containing NaCl for charge neutralization). **e** Typical molecular cluster and the non-bonded interactions involved in FFKY assemblies in detail. **f** Molecular structure of FG8. **g** The FG8 assemblies generated at $t = 200$ ns of MD simulation of 16 FG8 molecules in water (containing NaCl for charge neutralization). **h** Typical molecular cluster and the non-bonded interactions involved in FG8 assemblies in detail. The dashed purple lines denote the π-π stacking. The dashed blue lines denote the hydrogen bonds. The nitrogen atoms are labeled in blue. And the oxygen atoms are labeled in red. Source data are provided as a Source Data file.

incriminating tools for the microenvironment re-education, were also detected, i.e. ROS, iNOS and arginase-1 (Supplementary Fig. 10a, c)[16]. Major characteristics of PMN also emerged in lung as time went by, such as activation of fibroblasts (Supplementary Fig. 10e, f), angiogenesis (Supplementary Fig. 10g) and increase of vascular permeability (Supplementary Fig. 10h)[12]. All these data indicated that the in vivo PMN model had been well-established, reflecting the pathological process of PMN development. With the overall support provided by PMN, it would consequently accelerate and aggravate metastasis in vivo (Supplementary Fig. 13).

**FR17 administration interrupts the activation of fibroblast induced by tumor-derived factors**. According to previous researches on tumor metastasis[18] and our assessment on MCM-induced PMN mice model, the activation of the resident fibroblasts in distal tissues could be regarded as the tipping point of

the beginning of PMN establishing, raising the alarm about laying the foundations of the potential metastasis. It is reported that tumor-educated fibroblasts would serve to construct a tumor-supportive host stroma via STAT3 signaling[17] and JNK signaling[18] pathways, promoting ECM degradation and recon-struction, inducing angiogenic and pro-inflammatory response of endothelial cells, recruiting VLA-4[+] bone marrow-derived cells by localized FN deposition for niche formation[19]. Indeed, lysyl oxidase (LOX) cross-linked collagen surrounding pre-metastasis associated fibroblasts attracts CD11b[+] myeloid cells invasion in destination organs, which corresponds to metastatic efficiency[47].

Here in a cell model simulating the impact of secreted factors derived from primary tumor on lung fibroblasts in vitro (Fig. 3b), FR17 cultivation along with MCM stimulation showed an interruption on tumor-derived factors irritating mouse lung fibroblasts (MLF) (Fig. 3a). The treatment with FR17 or sFD17, which would form peptide nano-blanket assembled by the

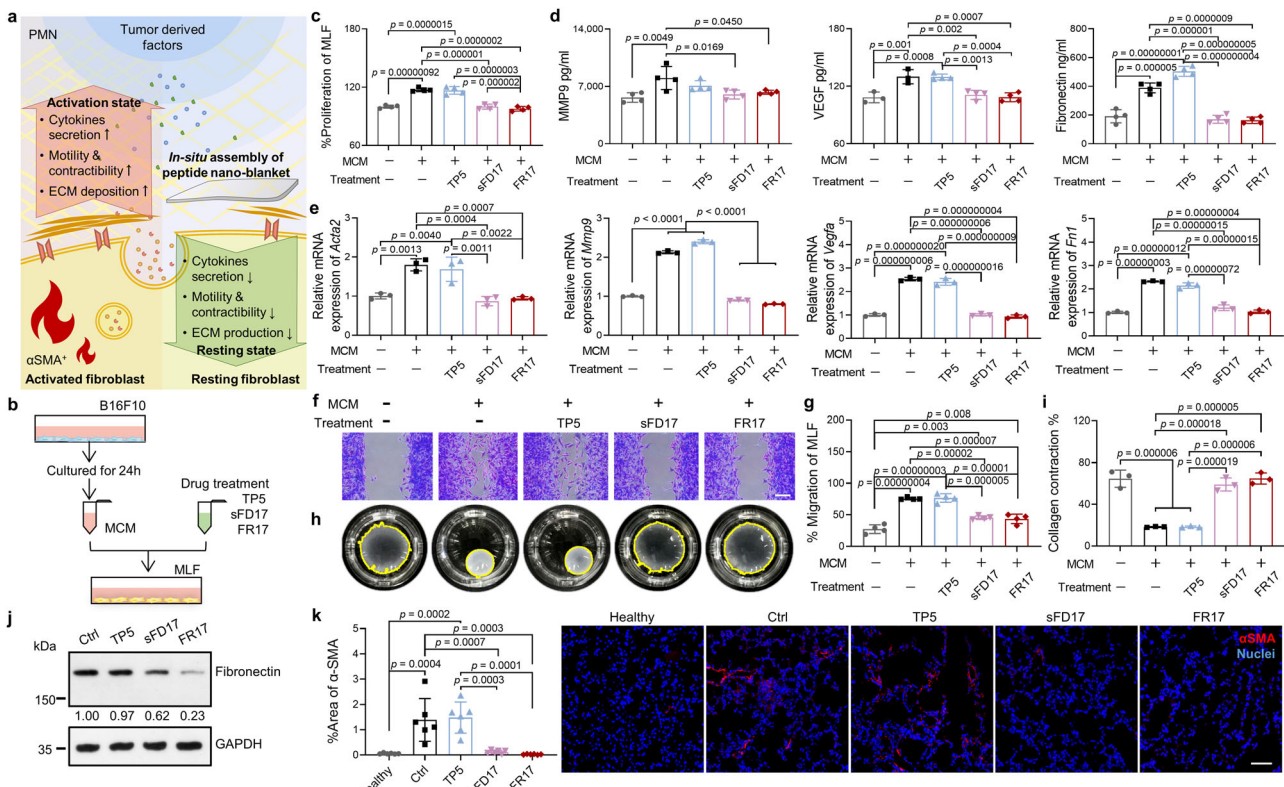

**Fig. 3 FR17 interrupts the activation of fibroblast induced by tumor-derived factors. a** Schematic drawing illustrates the in-situ assembled peptide nano-blanket interrupts tumor-derived factors-induced activation of fibroblast. **b** Illustration of the protocol of MCM stimulation and peptide treatment on MLF in vitro. **c** Cell proliferation of MLF after MCM stimulation and peptide treatment. $n = 4$. **d** Secretion of MMP9 ($n = 4$), VEGF ($n = 4$ for peptide treated groups and 3 for the control groups) and fibronectin ($n = 4$) in the culture media of MLF after stimulated by MCM, treated with or without peptide. **e** qPCR analysis of *Acta2* (i.e., *αSma*), *Mmp9*, *Vegfa*, *Fibronectin1* (*Fn1*) expression in MLF after MCM stimulation and peptide treatment. $n = 3$. For *Mmp9*: MCM free vs. MCM, $p = 0.000000000011$; MCM free vs. TP5, $p = 0.00000000000056$; MCM vs. sFD17, $p = 0.0000000000037$; MCM vs. FR17, $p = 0.0000000000013$; TP5 vs. sFD17, $p = 0.00000000000017$; TP5 vs. FR17, $p = 0.000000000000071$. **f, g** Migration assay to evaluate MLF migration ability after MCM stimulation and peptide treatment. Scale bar $= 100\ \mu m$. $n = 4$. **h, i** Collagen gel contract assay to evaluate contracting function of MLF after MCM stimulation and peptide treatment. $n = 3$. **j** Expression level of fibronectin in the lung harvested from the PMN model mice administrated with different peptides on day 10. **k** Representative images and semi-quantification of αSMA⁺ fibroblasts in the lung harvested from mice administrated with different peptides on day 10. Scale bar $= 50\ \mu m$. $n = 6$. Data are presented as mean ± SD for all column charts. One-way ANOVA followed by Tukey's multiple comparisons test was employed for statistical evaluation for **c–e**, **g**, **i**, **k**. Source data are provided as a Source Data file.

monomer FG8 that was released by enzyme-induced cleavage, successfully prevented the activation of fibroblasts by MCM (Supplementary Fig. 14). The inhibition on the expression of pathologic fibroblast biomarker alpha smooth muscle actin (αSMA) as well as the fibroblasts proliferation and cellular biofunctions (Fig. 3c–i) also substantiated the blocking of lung fibroblasts activity by FR17 or the scrambled control sFD17. The reverse of the increase in cytokines secretion, like MMP9, VEGF, FN, was determined by ELISA kit (Fig. 3d). The qPCR results further suggested that the peptide-assemblies might exert biological regulatory effect on cells while not just acting like a physical shield to wrap on the surface of cells (Fig. 3e). Besides, when irritated by tumor-derived factors to present the activated αSMA⁺ phenotype, the migration ability as well as the collagen gel contracting function of fibroblasts got promoted. By contrast, cultivation with FR17 minified these functional changes to a large extent (Fig. 3f–i). Administration of FR17 to the in vivo PMN model also gave great relief to the activation of the fibroblasts in lung (Fig. 3k). What's more, the above data also suggested that the "flame-retarding" effect on fibroblast activation was almost completely contributed by peptide nano-blanket on account of the similar outcomes of the scrambled control sFD17 to that of FR17 treatment.

The stromal microenvironment, apart from fibroblasts, consisting of ECM and vasculature, could also be re-educated by tumor-derived factors[48]. For ECM would have gone through remodeling to rebuild a supportive niche for tumor colonization in PMN model, Western blot analysis, Masson staining and Sirius Red staining revealed the down-regulation in the expression of FN (Fig. 3j) and collagen (Supplementary Fig. 15a–c) in the lungs of FR17 group and sFD17 group, which would have likely been over-expressed by the activated fibroblasts otherwise. Besides, versican expression in the lung was also lower in the FR17 or sFD17 intervened groups than that of PMN control group and TP5 treated group (Supplementary Fig. 15d), and the elevated expression of which has been reported to contribute to angiogenesis in tumor[49].

**FR17 protects fibroblasts from activation to inhibit vascular leakage and angiogenesis.** The activated fibroblasts in primary tumor, also known as cancer-associated fibroblasts (CAFs), have been reported to produce proangiogenic factors, such as VEGF, so as to promote tumor angiogenesis[17]. What's more, the secretion of MMPs by CAFs induced tumor vascular leakage, exacerbating MDSCs and tumor cells intravasation into the vascular system. Therefore, we assumed that, as an important

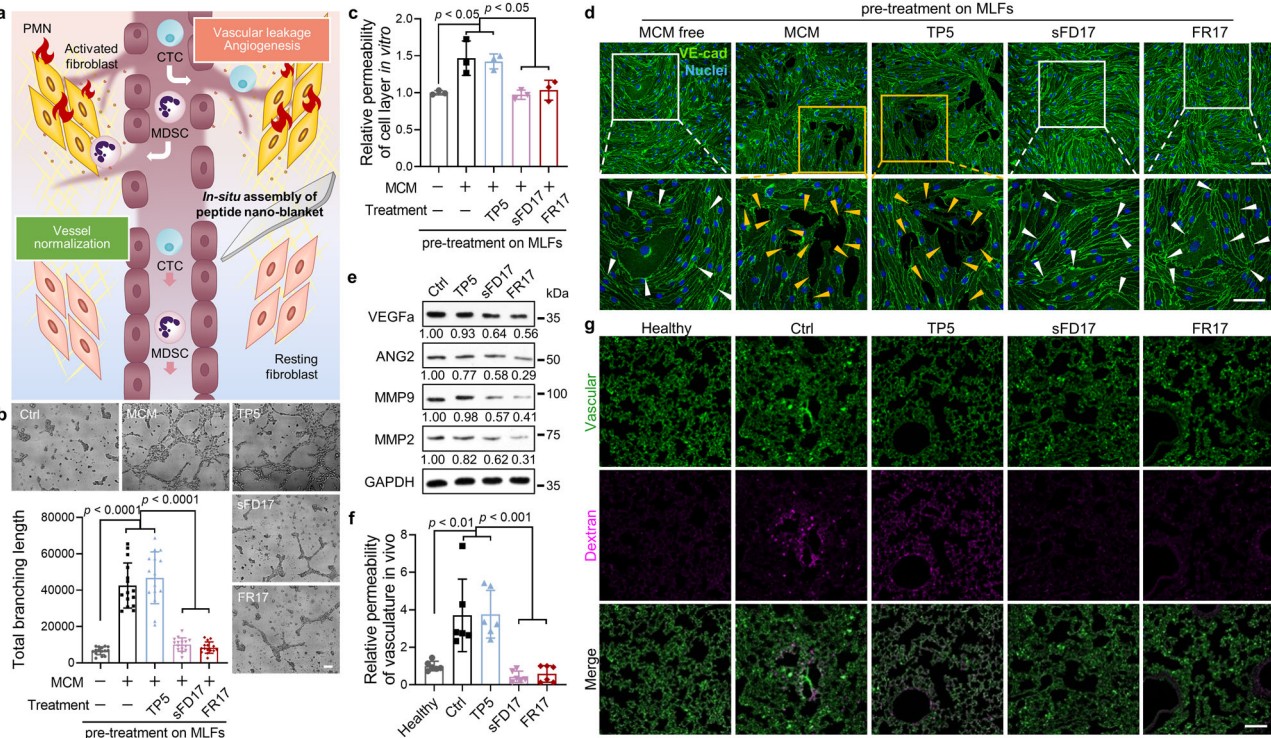

**Fig. 4 FR17 administration protects fibroblasts from activation to inhibit vascular leakage and angiogenesis. a** Illustration of the inhibition of vascular leakage and angiogenesis via the peptide nano-blanket protection on fibroblasts from tumor re-education. **b** Tube forming assay evaluating endothelial cells' neovascularization when treated with conditional FCM. Scale bar = 100 μm. The semi-quantification was calculated from five random fields imaged per well from triplicate wells per group via ImageJ. Data are presented as mean ± SD. **c** Permeability of the endothelial cell layer in vitro when co-cultured with MCM and peptides pre-treated MLF. Data are presented as mean ± SD. n = 3. MCM free vs. MCM, p = 0.0104; MCM free vs. TP5, p = 0.0192; MCM vs. sFD17, p = 0.0072; MCM vs. FR17, p = 0.0168; TP5 vs. sFD17, p = 0.0132; TP5 vs. FR17, p = 0.0314. **d** The endothelial cell monolayer's integrality after cultivated with conditional FCM, indicated by VE-cadherin. The white box and arrows indicate the adherens junction between endothelial cells. While the yellow box and arrows indicate the disruption of adherens junction. Scale bar = 50 μm. **e** Expression level of VEGFa, ANG2, MMP9 and MMP2 in the PMN lungs harvested from mice administrated with different peptides on day 10. **f, g** Vascular permeability in the PMN lung after peptide administration. Scale bar = 100 μm. Data are presented as mean ± SD. n = 6. Healthy vs. Ctrl, p = 0.0016; Healthy vs. TP5, p = 0.0013; Ctrl vs. sFD17, p = 0.0002; Ctrl vs. FR17, p = 0.0003; TP5 vs. sFD17, p = 0.0001; TP5 vs. FR17, p = 0.002. One-way ANOVA followed by Tukey's multiple comparisons test was employed for statistical evaluation for **b**, **c**, **f**. Source data are provided as a Source Data file.

participator and vanguard in the construction of PMN, the activated fibroblast may also contribute to angiogenesis and the increased vascular permeability in PMN (Fig. 4a). Further exploration was carried out on mice vascular endothelial cells to find out whether fibroblasts promote angiogenesis and vascular permeability during the arousement induced by MCM (as sketched in Supplementary Fig. 16a). The tube forming assay demonstrated the proangiogenic capability of fibroblasts activated by MCM (Fig. 4b). Meanwhile, the transwell permeability assay (as illustrated in Supplementary Fig. 16b) to mimic the inner layer of blood vessel in vitro verified that the fibroblast-conditioned medium (FCM) collected from activated MLF increased vascular permeability (Fig. 4c), indicated directly by the dismission of endothelial adherence junctions mediated by vascular endothelial cadherin, VE-cadherin (Fig. 4d).

For all experiments conducted on endothelial cells, the different conditional FCM was obtained from fibroblasts pre-treated with FR17, sFD17 or TP5 on interrupting MCM stimulation separately. The conditional FCM pre-treated with FR17 or sFD17, rather than TP5, offset the increase in bEnd3 cell proliferation induced by the indirect stimulation of MCM (Supplementary Fig. 16c). The acceleration in neovascularization and the disruption on endothelial cell-cell connection to cause vascular leakage were made up by indirect FR17 or sFD17 treatment on MLF as indicated in Fig. 4b, c.

To examine the influence of FR17 intervention on the expression level of the proangiogenic factors and vascular remodeling enzymes in pulmonary PMN in general, Western blot analysis was carried out. When compared to PMN control group, the increased expression of VEGFa, ANG2, MMP9, MMP2 was reversed by FR17 treatment almost back to normal levels (Fig. 4e and Supplementary Fig. 17). In addition, the scrambled control sFD17, which does not possess the drug-bioactivity of TP5, exhibited similar inhibition result as FR17 while TP5 did not, suggesting the protective effect was contributed by the peptide-assemblies formed by the enzyme-activatable self-assembled monomer. Vascular leakage assay on PMN model in vivo verified that FR17 or sFD17 administration attenuated the enhancement of pulmonary vascular permeability (Fig. 4f, g).

Above data confirmed that FR17 can be employed as an in-situ spontaneously-assembled "flame-retarding blanket" to beat out the "flames" on fibroblasts caused by tumor-derived factors in PMN, preventing pro-metastatic angiogenesis and vascular destabilization.

**FR17 administration impedes MDSC recruitment to pulmonary PMN and modulates the immuno-microenvironment.** Given the fact that MDSC occupies a vital position in PMN

construction and metastasis formation, MDSC has attracted wide concerns in recent years. It's firstly reported by David Lyden's team that VEGFR[+] myeloid progenitor cells are recruited and formed cell clusters to the sites highly expressing FN, which is most likely produced by resident fibroblasts, in the distal tissues prior to the arrival of tumor cells[14]. The recruitment of MDSC was found to be related to the enhancement of ECM production and remodeling in PMN, including cross-linking collagen by LOX secreted by tumor cells[47], FN deposition[50], periostin (POSTN)-enrichment[51]. As MDSC plays a crucial role in developing the immunosuppressive and inflammatory microenvironment, some pioneers have demonstrated that blocking the recruitment of MDSC could be a promising strategy to suppress early metastasis by preventing the development of breeding ground for tumor[52–55].

In our PMN model, the accumulation of MDSC in lung was decreased in FR17 and sFD17 groups, revealed by immunofluorescent staining (Fig. 5a) and flow cytometry analysis on day 10 (Fig. 5b). Moreover, focal enrichment areas of FN and the co-localization of LOX attracted an increasing number of CD11b[+]Gr[+] MDSC in serial sections of the PMN lung comparing to healthy lung (Fig. 5a and Supplementary Fig. 18), consistent with the previous reports[47]. The immunofluorescent stains indicated that FR17 intervention successfully down-regulated the expression of LOX, FN and POSTN in the lung induced by tumor-derived factors to impede the construction of PMN, which probably resulted in the decrease in MDSC recruitment (Supplementary Fig. 18). To evaluate prevention of peptide administration on developing PMN immunosuppressive environment, protein level of TGF-β in PMN (Fig. 5c), the well-known immuno-modulator produced by MDSC to regulate the establishment of immunosuppressive tumor-supportive niche, as well as the pro-inflammatory cytokine IL-6 was detected (Fig. 5d). FR17 treatment successfully down-regulated TGF-β and IL-6 expression in lung, so as sFD17. In the meantime, TP5 administration exhibited partial abatement on the expression of these cytokines[56], suggesting both the enzyme-responsive-assembled peptide nano-blanket and the immunoregulatory agent TP5 facilitated the normalization of immunosuppressive and inflammatory environment of PMN.

To find out whether the impact on MDSC-induced immunosuppressive microenvironment in PMN was mainly contributed by TP5 or by the in-situ assembled peptide nano-blanket, cell transcriptomic analysis of MDSCs recruited to the lung was carried out on different treatment groups. Given the fact that CD11b[+]Ly6g[+] MDSC takes the majority (over 90%) of MDSC population in the lung of MCM-induced PMN model (Supplementary Fig. 11a, b), CD11b[+]Ly6g[+] MDSC were sorted on day 10 from pulmonary PMN after different treatments as the representative subset for further transcription analysis (Supplementary Fig. 19). GO enrichment analysis indicated that the regulation effect of FR17 might relate to the activation of immune response pathway, cytokine production involved in immune response, regulation of leukocyte and lymphocyte activation, leukocyte chemotaxis and migration (Fig. 5e). Relative enriched pathways provided possible comments on the underlying mechanisms, including: regulation on cytokine biosynthetic process, adaptive immune response based on somatic recombination immune receptors built from immunoglobulin super-family domains, interferon-γ-mediated signaling pathway and the impact on chemokine receptor binding as well as CXCR chemokine receptor binding (Supplementary Fig. 20a). From above, when compared with the differential gene enrichment pathway of sFD17 and TP5, we found that the regulation on immune cells chemotaxis and migration pathways was contributed by the in-situ peptide-assemblies, for the peptide nano-blanket would only form in FR17

or sFD17 treated lung while not in free TP5 treated group. And the Venn diagram and further enrichment analysis to put TP5 aside suggested these would correspond to the regulation on cell surface receptor signaling pathway by peptide-assemblies (Supplementary Fig. 20b, c). In addition, the regulation on leukocyte differentiation and T cell activation pathway was mainly contributed by TP5 (Supplementary Fig. 20d), which has been commonly accepted as one of the mechanisms for TP5 to exert immune regulation effect[57–59].

Moreover, the in-situ assembled peptide nano-blanket formed by FR17 inhibited tumor cells migration as well (Supplementary Fig. 21c), with hardly any inhibition on cell viability (Supplementary Fig. 21a).

**FR17 administration inhibits melanoma lung metastasis in vivo.** First, we evaluated the anti-metastasis efficacy of FR17 on MCM-induced lung metastasis model. C57BL/6 mice were pre-induced by MCM administration for 10 days to set up the tumor-supportive PMN in lung. Tumor cells were then injected through the tail vein on day 7 to simulate the CTC wandering through blood vessels and eventually settled down in pulmonary PMN. As illustrated in Fig. 6a, peptide subcutaneous administration started from day 3 and ended at 2 weeks-post lung tumor inoculation. The metastasis inhibition efficacy was evaluated in terms of the tumor nodule number on day 28. FR17 effectively suppressed tumorigenesis and the development of metastasis in lung, compared to both the control and TP5 groups (Fig. 6b–d). Moreover, the good outcomes of sFD17 treatment which was close to that of FR17 group emphasized the main role of nano-blanket assembled by the enzyme-activated released monomer in intervening PMN construction. Therefore, the in-situ assembled nano-blanket also played a part in inhibiting metastasis development and growth. Preliminary safety evaluation on these model mice at the end of the experiment revealed no severe safety concerns of FR17 administration (Supplementary Figs. 22 and 23). The systemic immune modulation effect of TP5 was reflected on the recovery of thymus shrinking as well (Supplementary Figs. 22d and 23). What's more, peptide administration prevented tumor cells infiltration into the spleen (Supplementary Fig. 23).

The anti-metastasis activity of FR17 was further verified on a post-surgery metastasis model[52] (illustrated by Fig. 6e), which is closer to the clinical treatment of resectable melanoma followed-up with adjuvant therapy for patients at high risk of recurrence. Here we compared FR17 therapy with the rising-star of checkpoint inhibitors, PD1 antibody[60,61], which was approved by FDA as adjuvant therapy on primary tumor excised with lymph node management or metastatic melanoma[62]. As demonstrated in Fig. 6f, the recurrence of lung metastasis was put off and the overall survival was greatly improved by FR17 therapy when compared with control (Fig. 6g–i). The median overall survival prolonged for 25.9% (from 29 days to 36.5 days), catching up with the outcomes of anti-PD1 treatment (from 29 days to 31.5 days). And the median lung metastasis-free survival prolonged from less than 19 days of both control group and anti-PD1 treated group to 23 days of FR17 treated group. At the first time-point for lung metastasis monitoring via bioluminescence imaging, lung metastasis has been observed in all of the 10 mice from the control group, 8 mice from anti-PD1 treated group while only 3 mice from FR17 treated group on day 19. The retard of the occurrence of lung metastasis after FR17 administration emphasized the importance of PMN intervention. What's more, the relapse rate of the primary tumor post-resection was reduced from 10/10 in the control group to 4/10 in FR17 treated group (Supplementary Fig. 24).

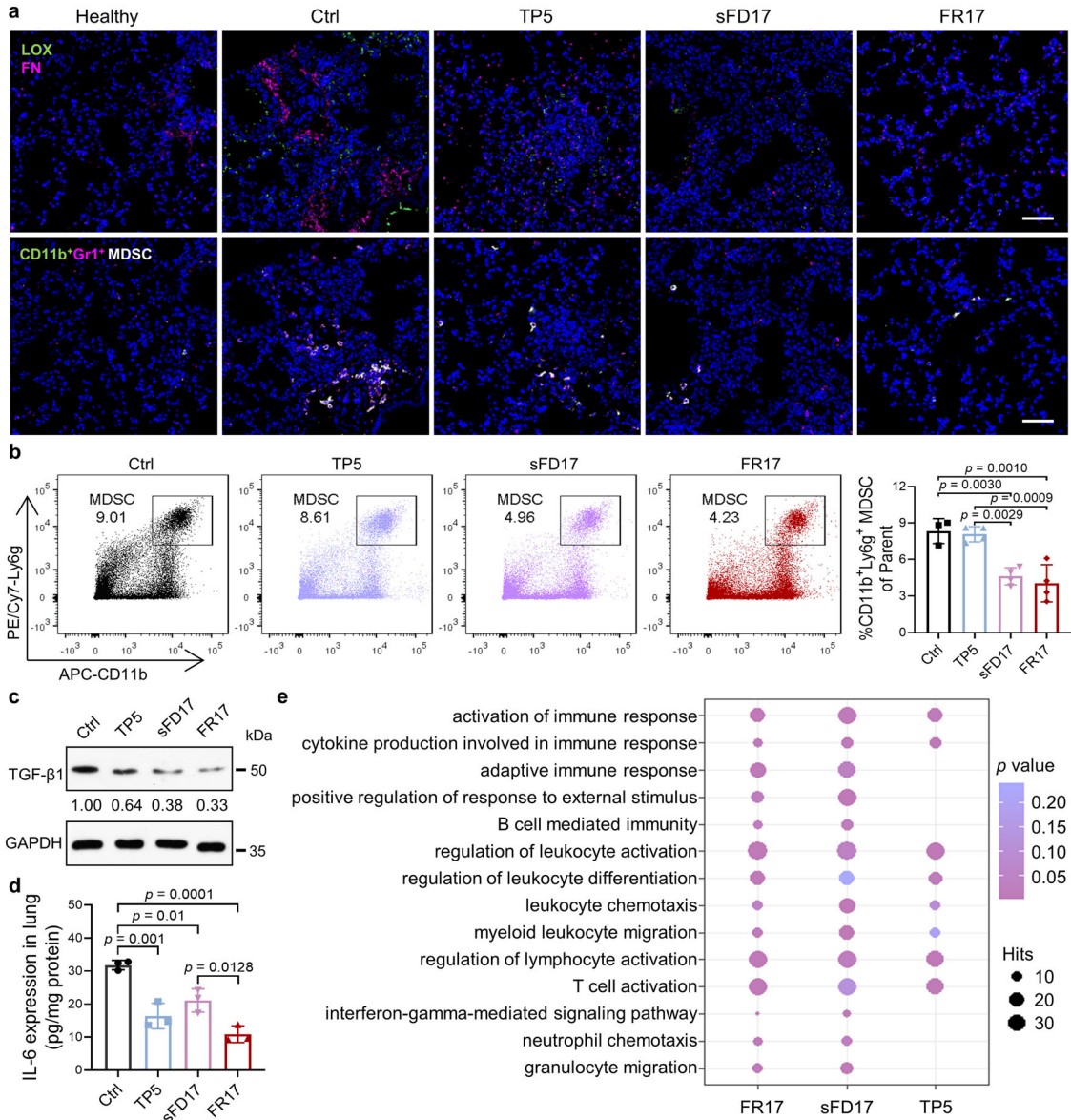

**Fig. 5 FR17 administration prevents MDSC recruitment, pausing the development of the immunosuppressive microenvironment in PMN. a** Serial sections of the lungs harvested from the model mice treated with different peptides. Serial sections show the distribution of lysyl oxidase (LOX) and Fibronectin (FN) and the co-location of CD11b+Gr1+ myeloid-derived suppressor cells (MDSC). Scale bar = 50 μm. **b** Recruitment of CD11b+Ly6g+ MDSC to the lungs of the model mice treated with different peptides on day 10. Data are presented as mean ± SD. $n = 4$ biologically independent mice for treatment groups, $n = 3$ for the control group. One-way ANOVA followed by Tukey's multiple comparisons test was employed for statistical evaluation. **c** Expression of TGF-β1 in the lung harvested from PMN model mice administrated with different peptides. **d** IL-6 expression in the lung tissue fluid collected from PMN model mice administrated with different peptides. Data are presented as mean ± SD. $n = 3$ biologically independent mice. One-way ANOVA followed by Tukey's multiple comparisons test was employed for statistical evaluation. **e** GO enrichment analysis of CD11b+Ly6g+ MDSC sorted from different treatment groups. RNA preparations were extracted from CD11b+Ly6g+ MDSCs sorted from lungs pooled from 10–12 mice per sample. The size of the dots corresponds to the number of genes per pathway, and the color indicates $p$ value. Statistical significance was considered at least at $p < 0.05$. Source data are provided as a Source Data file.

## Discussion

Though emerging technologies provided early detection of tumor metastasis[63] or even warning of high metastatic risk before the actual occurrence of metastasis[64], clinical treatment on metastasis prevention lags far behind. The contemporary therapeutic strategies against metastasis in clinic mainly focus on the later time period of macroscopic metastases development, or at least after the arrival and colonization of disseminated tumor cells to the distal organs, which have gained unsatisfied clinical outcomes[65–67]. As scientists further understanding the pathological process of tumor metastasis, the formation of PMN in distal organs arises by pro-metastatic

factors produced by primary tumor has drawn more and more attention[13].

With plenty of studies emphasized the vital role of MDSC in PMN[25,51–54], little has been explored on the contribution of stromal cells to re-educate the stromal microenvironment for the further development of PMN. Therefore, we would like to turn a spotlight on the pre-metastasis associated fibroblasts, the bugler who sounds the charge for the army of metastasis. Here, we have explored the magical retarding effect of the PMN micro-environment responsive-assembled peptide nano-blanket on fibroblasts activation, impeding PMN development. The enzyme-

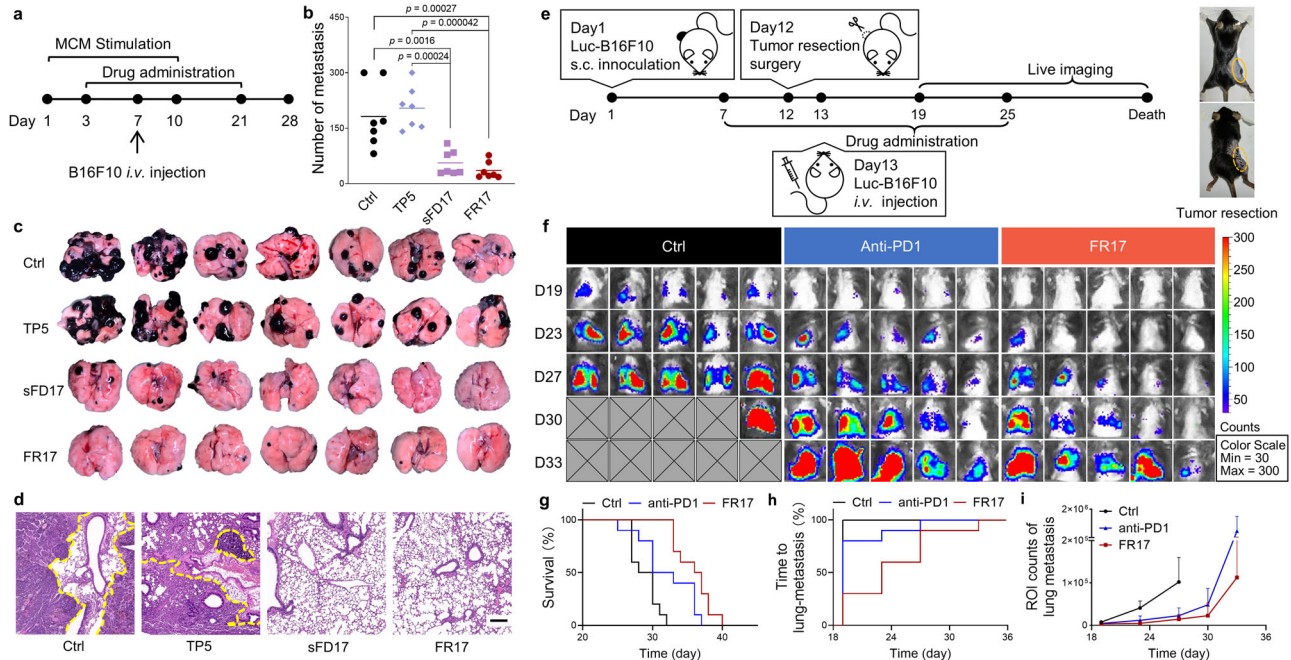

**Fig. 6 FR17 administration inhibits lung metastasis in vivo by retarding PMN formation. a** Schedule of MCM-induced lung metastasis model with peptide treatment. Peptide administration started from day 3 for metastasis prevention. **b** Number of the metastatic nodules in the lung of different treatment groups. $n = 7$. Data are presented as mean ± SD. One-way ANOVA followed by Tukey's multiple comparisons test was employed for statistical evaluation. **c** Images of the lung metastasis harvested on day 28 from different treatment groups. **d** Representative images of Hematoxylin and Eosin staining of lung sections from different treatment groups. The lung metastasis is circled by yellow dotted lines. Scale bar = 200 μm. **e** Schedule of post-surgery metastasis model. Peptide administration started from day 7 to 25. **f** Representative in vivo bioluminescent images of mice without treatment or treated with FR17 or anti-PD1. $n = 5$. The gray patches represent dead mice in the control group. **g** Survival curves of the mice treated with FR17 or anti-PD1 or without treatment. $n = 10$. **h** The cumulative incidence of new pulmonary metastases in the mice treated with FR17 or anti-PD1 or without treatment. $n = 10$. **i** Semi-quantification of the in vivo bioluminescent signals in the lungs of the mice treated with FR17 or anti-PD1 or without treatment. $n = 5$. Data are presented as mean ± SD. Source data are provided as a Source Data file.

activatable assembled peptide FR17 can be enzyme-cleaved to release the self-assembled monomer FG8 to construct a lamellar structure, which is named as peptide nano-blanket. Experiments demonstrated that FR17 administration not only beat out the "flame" set up on resident fibroblasts induced by tumor-derived factors, but also interrupted the subsequent PMN formation, including preventing pro-metastatic angiogenesis and vascular destabilization, and then intervening MDSCs' recruitment as well as their bio-functions. Astonishingly, when treated with sFD17, tumor-induced fibroblast activation was also impeded by peptide-assemblies without the assistance of TP5 so as to arrest the following pro-metastatic pathological process. This finding illustrated that the "flame-retarding" effect on fibroblast activation could be contributed by the drug-free peptide nano-blanket alone, presenting a broad application prospect of drug-free peptide-assemblies to regulate PMN microenvironment and to prevent tumor distant metastasis. This work also elucidated the important role of pre-metastasis associated fibroblast in the complex interactions between major participators in PMN formation and metastasis development, suggesting that reprogramming or intervention on the key juncture could make a big difference on fighting against tumor metastasis.

## Methods

**Ethical statement.** The animal study protocols (approval number: 21856) were approved by the Institutional Animal Care and Use Committee (IACUC) of Zhejiang University, and we have complied with all relevant ethical regulations.

*Characterization of FR17, sFD17 and FG8.* Peptide FR17, sFD17 and FG8 were synthesized via Fmoc solid-phase peptide synthesis technology by APeptide Shanghai. The molecule structure and amino acid sequence of peptide were

confirmed by mass and tandem mass spectra (Analyst TF v1.6 and PeakView software v1.2, AB Sciex). The purity of the peptides was analyzed by HPLC to be over 95% (Data provided in Supplementary Figs. 25–27).

*Mass and tandem mass spectrometry.* Mass spectrometry was conducted using AB Triple TOF 5600plus System (AB SCIEX, Framingham, USA), operated in the positive ion mode. File was created in Analyst TF (version 1.6). The exact mass calibration was performed automatically before each analysis using the Automated Calibration Delivery System. The data acquisition and analysis were performed by PeakView software (version 1.2, AB Sciex, USA).

*Liquid chromatography-tandem mass spectrometry.* To verify the enzymatic cleavage of FR17 and sFD17 to release FG8, peptide FR17 or sFD17 (200 μM, HBS buffer) was cultivated with pre-activated hMMP2 (200 ng/ml) for 10 min. The reaction was terminated by adding 3 volume of methanol and centrifuged at 13,000 rpm for 15 min to discard the protein precipitation. LC-MS/MS analyses were performed using an Agilent 6460 triple quadrupole mass spectrometer (Agilent Technologies, USA) equipped with an electrospray ionization source, operated in the positive ion multiple-reaction monitoring mode. Agilent MassHunter Workstation was used for data acquisition and processing. Chromatographic separation was carried out on a Zorbax SB C18 column (150 × 2.1 mm, 3.5 μm). Peptide alone was analyzed as standard control. The ion pairs to identify specific peptide sequence were auto-optimized by MassHunter Workstation Software, Optimizer for 6400 Series Triple Quadrupole (version B.09.00, Agilent Technologies, USA).

*Peptide self-assembly of FG8 in different pH conditions.* FG8 was dissolved in methanol, water, PBS buffer (pH 7.4) and PBS buffer (pH 6.8). Size distribution of peptide assemblies was measured by Zetasizer Nano ZS (Malvern Instruments). The assembly morphology of the assembled nano-blanket was observed with TEM (FEI Tecnai G2 spirit) for TEM image.

*Enzyme induced assembly of FR17 and sFD17.* Peptide FR17 or sFD17 was dissolved in HBS buffer (pH 7.4, containing 50 mM HEPES, 150 mM NaCl, 10 mM CaCl₂) at 500 μM. Activated hMMP2 was added to the peptide solution at the working concentration of 1 μg/ml. The solutions were then incubated in 37 ℃ air-

bath for 24 h. The size of the enzyme treated sample, which has formed the peptide nano-blanket, was measured by Zetasizer Nano ZS (software version 7.13, Malvern Instruments). The assembly morphology of the assembled nano-blanket was observed with TEM (FEI Tecnai G2 spirit) for TEM image and 200kv TEM (FEI Talos F200C) for Cryo-TEM image.

*All-atom molecular dynamics simulation.* The molecular structures of peptide FFKY and FG8 monomer were first constructed via AMBERTOOL based on the AMBER14SB force field. Dynamics simulation runs were performed utilizing Gromacs 2018.4 package[68]. System configurations were visualized using VMD software[69], and generated into images mainly employing GRACE and PyMOL v1.8.2.2 software. The simulation was performed in water boxes containing 16 FFKY or FG8 molecules. FFKY system was simulated containing NaCl to neutralize electric charge of the amidogen on the side-chain of Lys. Energy was minimized according to the steepest-descent method. Bond lengths were constrained by the LINCS algorithms. The non-bonded LJ interactions were cut off at 1.2 nm. Electrostatics was treated utilizing the Particle Mesh Ewald scheme. All production runs were simulated in the NPT ensemble using V-rescale coupling scheme with the temperature maintained at 298.15 K and parrinello-rahman coupling scheme with pressure kept at 1.0 bar and isotropic coupling type. The time constants for the pressure and temperature couplings were respectively set to 2.0 and 0.2 ps. Besides, the compressibility value was $4.5 \times 10^{-5}\,\mathrm{bar}^{-1}$. Periodic boundary conditions with a time step of 0.002 ps were adopted. Simulations were carried out for 200 ns and the structural coordinate information was recorded per 50 ps.

*Peptide self-assembly of TPE-FG8.* The AIE luminogens TPE was linked to the N-terminal of Phe of peptide, i.e. FFKY or FG8 or FR17, via glycine (synthesized via Fmoc solid-phase peptide synthesis technology by APeptide Shanghai, with >95% purity. Purity and MS identity are provided in Supplementary Figs. 28–31). For measurement of the AIE effect, TPE modified peptides were firstly dissolved in methanol as stock solution and then diluted with 99 volumes of water or PBS buffer (pH 6.8 or 7.4) to achieve self-assembly. The emission spectrum of final suspensions was detected using fluorescence spectrophotometer under the excitation wavelength at 405 nm. The microstructure of the assemblies was observed by Tecnai G2 spirit TEM (Thermo FEI, Czech Republic).

*Cell lines and cell culture.* B16F10 cells were purchased from Cell Bank of Chinese Academy of sciences (Cat. TCM36, Shanghai, China) which was originally obtained from the American Type Culture Collection (RRID:CVCL_0159, Manassas, USA) and cultured in Dulbecco's Modified Eagle Medium (DMEM, Cienry, China) containing 10% (v/v) fetal bovine serum (FBS, Gibco, Grand Island, USA) and penicillin-streptomycin Solution (100×, TBD, Tianjin, China) at 1% (v/v). And the luciferase transfected B16F10 (Luc-B16F10) was kindly gifted by Qichun Wei's lab, the Second Affiliated Hospital, Zhejiang University. The mouse lung fibroblasts (MLF) were purchased from iCell Bioscience Inc. (Cat. iCell-0033a, Shanghai, China) and cultured in Dulbecco's Modified Eagle Medium/Nutrient Mixture F-12, 1:1 mixture (DMEM/F12 medium, Multicell, Wisent Int., Canada) containing 10% (v/v) FBS (Gibco, Grand Island, USA) and penicillin-streptomycin Solution (100×, TBD, Tianjin, China) at 1% (v/v). The mouse endothelial cells bEnd3 were kindly gifted by Fuqiang Hu's lab, Institution of Pharmaceutics, College of Pharmaceutical Sciences, Zhejiang University, who originally purchased the cell from the Cell Bank of Chinese Academy of sciences (Cat. TCM40, Shanghai, China) which was originally obtained from the American Type Culture Collection (Cat. CRL-2299, RRID:CVCL_0170, Manassas, USA). The bEnd3 cells were cultured in the same medium as B16F10. Cells were cultured in a cell incubator containing 5% $CO_2$ at 37 °C and passaged when reach 80%-90% confluence. The MCM was obtained as follows: when B16F10 reached 70–80% confluence, washed with PBS and changed the medium to serum-free medium and incubated for 24 h. The cell supernatants were collected, centrifuged at $400 \times g$ for 10 min to discard the cell debris. The MLF-conditioned medium (FCM) was obtained as follows: MLFs were stimulated by MCM (supplemented with 10% (v/v) FBS) with or without peptide drugs (TP5, sFD17, FR17 at the concentration of 100 μM) for 48 h, the original medium was discard, then replaced with fresh complete medium after PBS rinsing to eliminate the direct influence of MCM and peptide drugs on the later experiment subject. After incubated for another 24 h, the cell supernatants were collected, centrifuged at $400 \times g$ for 10 min to discard the cell debris.

*Cell proliferation assays.* MLFs in rapid proliferation were plated in 96-well plates at the density of $4 \times 10^3$ per well and cultured overnight. Former media were removed and the cells were cultivated with 100 μl MCM supplemented with 10% (v/v) FBS, treated with or without peptide drugs (TP5, sFD17, FR17 at the concentration of 100 μM). Cells cultivated with fresh complete medium were set up as control. After incubation for 48 h, cell proliferation was measured using a Cell Counting Kit-8 assay (Cat. 13E02A60, Boster Biotech., China) according to the manufacturer's instructions. The optical density at 450 nm (OD450 nm) was measured using a multiwell plate reader (ELX800, BioTek, USA). Each group was repeated at least 4 times, and cell proliferation was presented as mean ± SD.

*The responsive-assembled peptide nano-blanket inhibiting fibroblasts activation.* MLFs were plated on round cover-slips at proper density and cultured overnight in 24-well plates. Former media were removed and replaced by conditional MCM, which contained with or without different peptides (TP5, sFD17, FR17 at the concentration of 100 μM). Fibroblasts cultured with fresh media were set as untreated control. After 24-h treatment, the fibroblasts attached on round cover-slips were solidified with glutaraldehyde (2.5%) overnight, fixed with osmium tetroxide, dehydrated and then coated with gold for SEM (Nova Nano 450, Thermo FEI, Czech Republic) observation.

*Cytokine secretion and gene expression of MLFs.* An equal number of MLFs in rapid proliferation were seeded in a 24-well plate per well, cultivated with MCM supplemented with 10% (v/v) FBS, treated with or without peptide drugs (TP5, sFD17, FR17 at the concentration of 100 μM) for 48 h in at least triplicate. Cells cultivated with fresh complete medium were set up as control. The cell supernatants were collected, centrifuged at $400 \times g$ for 5 min to discard cell debris. MMP9 ELISA kit (EK0466, Boster Biotech., China) and VEGF ELISA kit (EK0541, Boster Biotech., China) were employed to measure the secretion level of MMP9 and VEGF in the cell supernatant. An equal number of MLFs in rapid proliferation were seeded in 24-well plate cultivated with (serum-free) MCM, treated with or without peptide drugs (TP5, sFD17, FR17 at the concentration of 100 μM) for 48 h in quadruplicate. Cells cultivated with fresh DMEM/F12 medium were set up as control. The cell supernatants were collected, centrifuged at $300 \times g$ for 5 min to discard cell debris. The secretion of FN was measured by FN ELISA kit (EK0351, Boster Biotech., China) according to the manufacturer's instructions. Cells with different treatments were harvested for RT-qPCR analysis to determine the gene expression level of *M-Acta2* (Mouse αSMA), *M-Mmp9*, *M-Vegfa* and *M-Fn1*. The primer sequences are provided in Supplementary Table 1.

*Cell migration assays.* MLFs in rapid proliferation were plated in Culture-Inserts (2 Well, Ibidi, Germany) at the density of $1 \times 10^5$ in 70 μl per well and grew to confluence overnight. Culture-Inserts as well as the former medium were gently removed. Then the MLFs were cultivated with MCM supplemented with 10% (v/v) FBS, treated with or without peptide drugs (TP5, sFD17, FR17 at the concentration of 100 μM) in triplicate. Cells cultivated with fresh complete medium was set up as control. Pictures of the cell scratches were taken under the microscope at 0 h. After incubated for 24 h, MLFs were fixed and stained with crystal violet. Pictures were taken and analyzed with ImageJ (version 1.51j8).

*Collagen gel contraction assay.* An equal number of MLFs seeded in 6 cm dishes were cultivated with MCM supplemented with 10% (v/v) FBS, treated with or without peptide drugs (TP5, sFD17, FR17 at the concentration of 100 μM) for 48 h. Cells cultivated with fresh complete medium was set up as control. The pre-treated MLFs were digested and re-suspended at the density of $2 \times 10^6$ per ml, kept on ice for later use. The neutral collagen solution was prepared as follows: 224 μl type I collagen gel (3 mg/ml, Cat. C8062, Solarbio, China) was quickly mixed with 100 μl conditional medium and 8 μl NaOH (0.1 N) on ice. Then 300 μl of the pre-treated MLFs suspension was added to the collagen solution on ice immediately. And the neutral cell-collagen mixture was added to 48-well plates 200 μl per well in triplicate and allowed to solidify for 45 min at room temperature. After incubated at 37 °C for 12 h, the gels were photographed. ImageJ software (version 1.51j8) was used to measure gel area and evaluate contraction. Gel contraction was assessed as the ratio of the gel area to the area of the well.

*Peptides inhibiting MLF stimulation on the proliferation of bEnd3 cells.* bEnd3 cells in rapid proliferation were plated in 96-well plates at proper density and cultured overnight. Former media were removed and replaced by conditional FCM, which was obtained from MLFs stimulated by MCM with or without the treatment of different peptides (TP5, sFD17, FR17 at the concentration of 100 μM) as illustrated above. Cell cultivated with normal FCM was set up as control. Cell proliferation was measured after incubated for 48 h using a Cell Counting Kit-8 (CCK8) assay (Cat. 13E02A60, Boster BioTech., China), OD450 nm was read by a multiwell plate reader (ELX800, Biotek, USA).

*Tube forming assay.* Mouse endothelial bEnd3 cells seeded in 6 cm dishes were cultivated with conditional FCM (from MLF previously stimulated by MCM with or without TP5, sFD17 or FR17 peptide as illustrated in Supplementary Fig. 16a) for 24 h. The cells were harvested and seeded in 48-well plates pre-coated with Matrigel (Cat. 356230, BD, USA) in triplicate for each group. After 6 h incubation, five visual fields were randomly chosen from each well and photographed by microscope (CKX53, OLYMPUS, Japan). The tube forming results were analyzed by ImageJ (version 1.51j8).

*Transwell permeability assay.* An equal number of MLFs seeded in 6 cm dishes were cultivated with MCM supplemented with 10% (v/v) FBS, treated with or without peptide drugs (TP5, sFD17, FR17 at the concentration of 100 μM) for 48 h. Cells cultivated with fresh complete medium were set up as control. The pre-treated MLFs were digested and seeded at the lower well of a 24-well transwell plate at $2.5 \times 10^5$ cells per well. Each group was triplicate. Then, 7000 Mouse endothelial bEnd3 cells were seeded on a 0.4 μm Transwell insert (Cat. 3413, Corning Costar, USA) above the top of the well until grown to confluence. Rhodamine B-dextran

(70 kDa, Cat. R9379, Sigma-Aldrich, USA) was added to the upper insert on the endothelial cell layer. After 1 h incubation, the translocation of Rhodamine B-dextran from the insert to the lower well passing through the endothelial cell layer was measured by a microplate reader (SPARK, TECAN, Switzerland) at an excitation/emission wavelength of 540/625 nm. The relative permeability of the cell layer was normalized by diving the fluorescence signals of the treatment groups by the control group.

*Integrality of the endothelial cell monolayer.* Mouse endothelial bEnd3 cells grown in 35-mm confocal dishes to 100% confluence were treated with conditional FCM, which was obtained from MLFs stimulated by MCM with or without the treatment of different peptides (TP5, sFD17, FR17 at the concentration of 100 μM) as illustrated above. After incubated with conditional FCM for 24 h, the single endothelial cell layer was gently washed with PBS and fixed with 4% paraformaldehyde for 15 min, permeabilized with 0.2% Triton X-100 for 15 min and blocked with 2% bovine serum albumin (BSA) for another 15 min. Cells were incubated with anti-VE-cadherin antibody (1:1000, Cat. Ab205336, Abcam, UK) containing 0.2% BSA and 0.1% Triton X-100 in PBS at 4 °C overnight. Cells were washed with PBS for three times and incubated for 1 h with AF647 labeled goat anti-rabbit IgG (H + L) (1:100, Cat. 33113ES60, Yeasen, China). The nuclei were labeled by DAPI solution (ready-to-use) (Solarbio, China). Images were taken by confocal microscope (Leica, German).

*Proliferation and migration of tumor cells with FR17 treatment.* B16F10 cells in rapid proliferation were plated in 96-well plates at proper density and cultured overnight. FR17 at various concentrations (0–1000 μM) was cultivated with B16F10 for 24 or 72 h. Cell proliferation was measured by CCK8 assay (Cat. 13E02A60, Boster BioTech., China) as formerly introduced. Cell viability was normalized by the reads from untreated wells. For migration assay, tumor cells were plated in Culture-Inserts (2 Well, Ibidi, Germany) at proper density and grew to confluence overnight. Culture-Inserts as well as the former medium were gently removed. And the cells were cultivated with low, medium and high concentration of FR17 at 40, 200, 1000 μM in triplicate. Cell cultivated with fresh complete medium was set up as control. Pictures of the cell scratches were taken under the microscope at 0, 6, 12, 24 h respectively. After incubated for 24 h, cells were fixed and stained with crystal violet. Images were analyzed with ImageJ (version 1.51j8).

*Mice and animal models.* C57BL/6 mice (male, 5-week-old) purchased from Slaccas (Shanghai, China) were adaptive fed for more than 1 week for subsequent experiments. The animals were maintained under standard laboratory housing conditions (25 ± 1 °C, 50% relative humidity and 12 h/12 h dark/light cycle where foods and water can be reached freely). All the animal experiments were conducted following the guidelines which have been approved by the IACUC of Zhejiang University.

For MCM-induced lung metastasis model, MCM (300 μl per mice) was intraperitoneally injected to the mice (male, 6-week-old) for 10 consecutive days from day 1 to 10. On day 7, a tail vein injection of B16F10 or Luc-B16F10 cells ($1 \times 10^5$ per mice) was given to the mice. Lung metastasis was monitored twice a week by bioluminescence imaging (IVIS® Spectrum In Vivo Imaging System, PerkinElmer, USA) if viable. For bioluminescence imaging, mice were intraperitoneally injected with D-luciferin potassium salt (150 mg/kg, Gold Biotechnology, USA). The bioluminescence of pulmonary metastases was detected by Living Image® v4.3.1 10 min later.

For post-surgery metastasis model, $1 \times 10^6$ B16F10 cells were subcutaneously inoculated in the back of 30 C57BL/6 mice (male, 6-week-old) above the right hindlimb on day 1. On day 12, tumor resection surgery was conducted on mice to remove the entire tumor tissues as well as the skin cover the tumor under anesthesia. On the next day, $1 \times 10^5$ luciferase-expressing B16F10 cells were injected into mice through tail vein. The tumor recurrence and lung metastasis were monitored twice a week. The tumor volume and body weight were recorded twice a week. Tumor volume was calculated as $(width^2 \times length)/2$. Mice were immediately executed once the tumor volume exceeded permitted maximal tumor size (tumor average diameter > 15 mm) according to the ethical regulations provided by the the IACUC of Zhejiang University.

*Pre-metastatic niche study.* MCM (300 μl per mice) was intraperitoneally injected to the mice (male, 6-week-old) for 10 consecutive days. On day 7, a tail vein injection of B16F10 ($1 \times 10^5$ per mice, 12 mice) was given to the mice. On day 3, 6, 10, 13, mice were euthanized for cardiac perfusion and the lung tissues were collected for further analysis.

For Western blot analysis, total proteins from the lung tissues were extracted using Tissue Protein Extraction Reagent (T-PER™, Cat. 78510, Thermo Pierce, Thermo Scientific, USA) and quantified with a Bradford Protein Assay Kit (Cat. P0010, Beyotime, Beijing, China). Samples (60 μg) were separated on 10% or 8% SDS-PAGE gels, then transferred to PVDF nitrocellulose membrane (Cat. IPVH00010, Merck Millipore). Membranes were incubated with the appropriate primary antibodies in 3% BSA, including FN (1:500, Cat. ab2413, Abcam, UK), MMP9 (1:1000, Cat. ab38898, Abcam, UK), VEGFa (1:500, Cat. ab119, Abcam, UK), TGF-β1 (1:1000, Cat. ab179695, Abcam, UK), iNOS (1:1000, Cat. ab204017, Abcam, UK), Arginase 1 (1:1000, Cat. ab124917, Abcam, UK). Antibody against

GAPDH (1:10,000, Cat. ab181602, Abcam) was used as control. After incubation with appropriate secondary antibody Goat anti-Mouse IgG (H + L) (1:5000, Cat. 31160, Thermo Pierce) or Goat anti-Rabbit IgG (H + L) (1:5000, Cat. 31210, Thermo Pierce), the intensity of the immunoreactive proteins was stabilized by SuperSignal® West Dura Extended Duration Substrate (Cat. 34075, Thermo Pierce) and visualized on X-ray film.

For ELISA analysis, the lung tissues were ground and centrifuged at $10,000 \times g$ to gain supernatant to measure the MMP2 (Cat. OM457413, Omnimabs, USA) and ROS (Cat. OM641674, Omnimabs, USA) level.

For immunofluorescence staining, the left lung was fixed with 4% paraformaldehyde and 30% sucrose solution overnight, and embedded into paraffin and sliced into sections. The paraffin lung sections were deparaffinized and rehydrated, then stained with primary antibodies: αSMA (1:500, Cat. Ab7817, Abcam, UK), or CD34 (1:500, Cat. Ab81289, Abcam, UK). Secondary antibody Cy3 conjugated goat anti-rabbit IgG (1:500, Cat. 111-165-003, Jackson, USA) was utilized in 1:500 dilution and stained with DAPI before observation.

For flow cytometry analysis, lung tissues harvested from 5 PMN model mice were mechanically minced into 1–2 mm pieces using scissors and then dissociated into single-cell suspension at 37 °C on a shaker for 30 min by enzymes. The digesting solution contains 2 mg/ml collagenase I (Cat. BS163, BioSharp, Germany), 2 mg/ml collagenase II (Cat. BS164, BioSharp, Germany) and DNase I (Cat. KGF008, KeyGEN BioTech., China). Digestion was stopped by adding 2 volumes PBS and filtered through a 70 μM cell strainer (Cat. CSS013070, Jet BIOFIL®, China). The cell suspension was centrifuged at $400 \times g$ for 5 min to discard the supernatant. Cell precipitations were then resuspended in 5 ml RBC lysis buffer (Cat. R1010, Solarbio, China) and centrifuged again to discard the supernatant. The single-cell-suspensions washed with PBS and resuspended were incubated with FITC-antimouse-CD45 (1:500, Cat. 553079, BD, USA), PE-antimouse-NK1.1 (1:500, Cat. 108708, Biolegend, USA), PE-antimouse-CD3 (1:500, Cat. 100205, Biolegend, USA), PE-antimouse-TER119 (1:500, Cat. 116207, Biolegend, USA), PE-antimouse-CD19 (1:500, Cat. 152407, Biolegend, USA), APC-antimouse-CD11b (1:250, Cat. 101211, Biolegend, USA), BV605-antimouse-MHC II (1:150, Cat. 107639, Biolegend, USA), BB700-CD11c (Cat. 566505, BD, USA), BV421-antimouse-Ly6c (1:100, Cat. 562727, BD, USA) and PE/CF594-antimouse-Ly6g (1:150, Cat. 562700, BD, USA), BV711-antimouse-F4/80 (1:150, Cat. 123147, Biolegend, USA), PE/Cy7-antimouse-CD103 (1:100, Cat. 121426, Biolegend, USA) antibodies in 100 μl 1% BSA containing 50 μl BD Brilliant Stain Buffer for 30 min at 4 °C in dark. After centrifuged and washed with PBS, cell pellets were fixed and membrane were perforated with Fix/Perm Buffer (Cat. 562574, BD, USA). The cell pellets were then stained with BV650-antimouse-CD206 (1:50, Cat. 141723, Biolegend, USA) for 40 min in the dark at room temperature. After centrifuged and washed with PBS, the stained cell pellets were analyzed by BD Fortessa flow cytometry. The data were analyzed using FlowJo software v10.6.2.

To investigate the impact of PMN formation induced by MCM injection, metastasis development and mice survival were monitored on MCM-induced PMN model and on the mice (male, 6-week-old) that did not receive MCM injection but were inoculated directly with Luc-B16F10 cells ($1 \times 10^5$ per mice) on day 7. Lung metastasis was monitored twice a week by bioluminescence imaging (IVIS Spectrum, USA).

*STEM observation of the in situ assembly of peptide nano-blanket in vivo.* FR17 was subcutaneously administrated to PMN mice (day 10) at the dose of 100 μM/kg. In total, 12 h-post peptide administration, mice were euthanized for cardiac perfusion and lung tissues were collected and fixed with 2.5% glutaraldehyde, embedded and sliced for further STEM observation.

*The aggregation-induced emission effect of the in situ assembly of peptide nano-blanket in vivo.* TPE modified FR17 was subcutaneously administrated to PMN mice (day 10) or healthy mice at the dose of 100 μM/kg. In total, 12 h-post peptide administration, mice were euthanized for cardiac perfusion and major organs were collected for frozen sections. The frozen sections were observed under high-resolution laser confocal microscopy (Leica, Germany) excited by 405 nm laser.

**Influence of peptide interference on mice PMN.** Twelve PMN model mice were randomly divided into four groups, namely control, TP5, sFD17 and FR17. Peptides, including TP5, sFD17 and FR17, were administrated separately to the mice subcutaneously from day 3 at 40 μM/kg/day. On day 10, mice were euthanized for cardiac perfusion and the lung tissues were collected for further analysis.

The lung tissues harvested from different groups were fixed, embedded into paraffin and sliced into sections. To visualize the activation of lung fibroblasts and angiogenesis in pulmonary PMN, the paraffin lung sections after deparaffinization and rehydration, were stained with primary antibodies: αSMA (1:500, Cat. ab7817, Abcam, UK), Vimentin (1:2000, Cat. Cy5134, Abways, China) or CD34 (1:500, Cat. ab81289, Abcam, UK). Secondary antibody Cy3 conjugated goat anti-rabbit IgG (Cat. 111-165-003, Jackson, USA) was utilized in 1:500 dilution and stained with DAPI before observation. To visualize the ECM environment alteration in pulmonary PMN, the lung sections were stained with appropriate primary antibodies: MMP2 (1:200, Cat. 10373-2-ap, PTG, USA), or MMP9 (1:1000, Cat. ab228402, Abcam, UK), Secondary antibody Cy3 conjugated goat anti-rabbit IgG (1:500, Cat. 111-165-003, Jackson, USA) and DAPI. To visualize the collagen

deposition in pulmonary PMN, Masson's trichrome staining of the lung sections were imaged and analyzed by ImageJ (version 1.51j8) to calculate the collagen volume fraction by dividing the blue collagen area by total tissue area. And the Sirius Red Staining was also carried out and the sections were visualized under the polarizing microscope (Nikon Eclipse Ci). To investigate the recruitment of MDSC in PMN, serial sections of lung tissues were stained with periostin (1:200, Cat. 19899-1-AP, PTG, USA), or co-stained with LOX (1:200, Cat. ab174316, Abcam, UK) and FN (1:200, Cat. ab92572, Abcam, UK) antibody, or CD11b (1:2000, Cat. ab133357, Abcam, UK) and Gr-1 (1:200, Cat. ab25377, Abcam, UK) antibody separately. Secondary antibody Cy3 conjugated goat anti-rabbit IgG (1:500, Cat. 111-165-003, Jackson, USA), goat anti-rabbit IgG conjugated to HRP (1:2000, Cat. ab6721, Abcam, UK) and fluorescent TSA-488 (1:200, Wuhan Pinuofei, China) were applied according to Tyramide Signal Amplification technology. The stained sections were observed and imaged under the confocal microscope (ECLIPSE, Nikon). Images were analyzed by ImageJ (version 1.51j8) if necessary.

To investigate the alteration of protein expression level in PMN, Western blot or ELISA experiments were carried out. For Western blot assay, protein samples were extracted separately from 3 independent mice from each group. Total proteins from the lung tissues were extracted using Tissue Protein Extraction Reagent (T-PER™, Cat. 78510, Thermo Pierce, Thermo Scientific, USA) and quantified with a Bradford Protein Assay Kit (Cat. P0010, Beyotime, Beijing, China). Samples (60 μg) were separated on 10% or 8% SDS-PAGE gels, then transferred to PVDF nitrocellulose membrane (Cat. IPVH00010, Merck Millipore). Membranes were incubated with the appropriate primary antibodies in 3% BSA, including FN (1:500, Cat. ab2413, Abcam, UK), Versican (1:1000, Cat. ab270445, Abcam, UK), VEGFa (1:500, Cat. ab119, Abcam, UK), ANG2 (1:500, Cat. ab155106, Abcam, UK), MMP9 (1:1000, Cat. ab38898, Abcam, UK), MMP2 (1:500, Cat. ab97779, Abcam, UK), TGF-β1 (1:1000, Cat. ab179695, Abcam, UK). Antibody against GAPDH (1:10,000, Cat. ab181602, Abcam) was used as control. After incubation with secondary antibody Goat anti-Mouse IgG (H + L) (1:5000, Cat. 31160, Thermo Pierce) or Goat anti-Rabbit IgG (H + L) (1:5000, Cat. 31210, Thermo Pierce), the intensity of the immunoreactive proteins was stabilized by SuperSignal® West Dura Extended Duration Substrate (Cat. 34075, Thermo Pierce) and visualized on X-ray film. For ELISA assay, lung tissues were ground and centrifuged to gain supernatant to measure the IL-6 (Cat. EK0411, Boster, China) level.

*In vivo vascular permeability assay*. MCM (300 μl per mice) was intraperitoneally injected to 12 mice (male, 6-week-old) for 7 consecutive days. And the mice were randomly divided into four groups, namely control, TP5, sFD17 and FR17. Peptides, including TP5, sFD17 and FR17, were administrated separately to the mice subcutaneously from day 3 at 40 μM/kg/day. On day 7, 100 mg/kg Rhodamine B-dextran (70 kDa, Cat. R9379, Sigma-Aldrich, USA) was intravenously injected to the mice. After 3 h, mice were injected with FITC-lectin (Cat. L0770, Sigma-Aldrich, USA) at 10 mg/kg through the tail vein. Ten minutes later, each mouse was anesthetized and transcardiac perfused with 20 ml saline to remove the excess dye and followed by 5 ml of 4% formaldehyde. The lung tissues were formaldehyde-fixed and cryo-sectioned. Slices were observed and imaged by fluorescence microscopy (Leica, German) for vascular leakage. There were three mice in each group and five visual fields were randomly chosen and images were taken by Leica Application Suite X for each section. The relative vascular permeability was analyzed by dividing the dye leakage of each group by healthy control.

*Recruitment of MDSC to PMN*. On day 10 of PMN mice model, lung tissues were harvested from different treatment groups and mechanically minced and digested to obtain the single-cell-suspensions as described above. The single-cell suspensions washed with PBS and resuspended were incubated with APC-antimouse-CD11b (1:250, Cat. 101211, Biolegend, USA) and PE/Cy7-antimouse-Ly6g (1:150, Cat. 127617,Biolegend, USA) antibodies in 100 μl 1% BSA for 30 min at 4 °C in dark. After centrifuged at $400 \times g$ and washed with PBS, cell pellets were analyzed by BD Fortessa flow cytometry. The data were analyzed using FlowJo software v10.6.2.

*mRNA sequencing of CD11b⁺Ly6g⁺ MDSC recruited to PMN*. On day 10 of PMN mice model, lung tissues were harvested from different treatment groups and digested into single cells as introduced as above. The single-cell suspensions after washing with PBS and re-suspension were incubated with APC-antimouse-CD11b (1:250, Cat. 101211, Biolegend, USA) and PE/Cy7-antimouse-Ly6g (1:150, Cat. 127617, Biolegend, USA) antibodies in 100 μl 1% BSA for 30 min at 4 °C in the dark. After centrifuged at $400 \times g$ and washed with PBS, CD11b⁺Ly6g⁺ MDSCs were sorted from the PMN lungs of 10–12 individual mice from each group per sample by FACS (Beckman moflo Astrios EQ). Total RNA was extracted by TRIzol for cDNA preamplification using the NEBNext® Ultra™ Directional RNA Library Prep Kit for Illumina®, then analyzed using Qubit2.0 Fluorometer, Agilent 2100 bioanalyzer and qRT-PCR. Significantly enriched gene sets were defined as p values < 0.05 comparing to the control group.

*Inhibition of tumor metastasis on MCM-induced PMN lung metastasis model*. The PMN models were established as introduced above, 28 PMN model mice were randomly divided into four groups ($n = 7$), namely control, TP5, sFD17 and FR17.

From day 3 to 21, mice from different groups were subcutaneously administrated with saline, TP5 (40 μM/kg per day), sFD17 (40 μM/kg per day) and FR17 (40 μM/kg per day) separately. Body weight was recorded every 3 days. On day 20, blood was collected from the submarginal ocular venous plexus under anesthesia for blood tests including complete blood count, alanine aminotransferase, aspartate aminotransferase, blood urea nitrogen and serum creatinine (CREA). On day 28, mice were euthanized and major organs, such as heart, liver, spleen, lung, kidney and thymus were collected after cardiac perfusion. Lung tumor nodules were then counted under the stereo microscope. The major organs were fixed, embedded into paraffin for Hematoxylin and Eosin staining. The thymus coefficient was calculated as the thymus weight divided by the body weight and the spleen coefficient was calculated as the spleen weight divided by the body weight.

*Inhibition of tumor metastasis post-surgery*. The post-surgery metastasis model was established as illustrated above. Thirty C57BL/6 mice (male, 6-week-old) were randomly divided into three groups, namely control, anti-PD1 and FR17. For FR17 treatment, peptide was subcutaneously administrated to the mice from day 7 to day 25 at the dose of 40 μM/kg per day. For anti-PD1 treatment, 100 mg anti-PD1 (Cat. BE0146, Bio X Cell, USA) was given by i.p. injection twice per week starting from day 3 post tumor resection and given two times per week from day 15 to day 25 for a total of four times. Lung metastasis was monitored twice a week by bioluminescence imaging (IVIS Spectrum, USA) until death. The recurrence of the excised subcutaneous tumor was closely monitored and tumor volume was measured and calculated following the ellipsoid volume formula: $(width^2 \times length)/2$.

**Statistics and reproducibility**. Statistical analysis was performed using GraphPad Prism 8.0.1 (GraphPad Software, CA, USA). Data were presented as means ± SD. Statistical evaluation of differences between two experimental groups was performed by two-tailed $t$ test. Statistical evaluation of differences between more than three experimental groups was performed by one-way ANOVA followed by Tukey's multiple comparisons test. Statistical significance was considered at least at $p < 0.05$. Representative images including micrographs and scans of blots such as Figs. 2a, 3j, 4e, 5a, c, 6d and Supplementary Figs. 4a, 5c, 6c–e, 7, 8, 10a, g, 14, 15, 17, 23 were representative for at least three independent experiments with similar results.

**Reporting summary**. Further information on research design is available in the Nature Research Reporting Summary linked to this article.

## Data availability

The mRNA-seq datasets generated in this study have been deposited in the NCBI GEO database under accession code GSE181898. All other data generated in this study are provided in the article, Supplementary Information files, Source Data file, or are available from the corresponding authors upon reasonable request. Source data are provided with this paper.

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

## Acknowledgements

This work was supported by the National Natural Science Foundation of China (Nos. 81673022, M.H.) and Ten-thousand Talents Program of Zhejiang Province (2018R52049, J.G.). We thank Professor Yu Kang at College of Pharmaceutical Sciences, Zhejiang University for guidance on molecular dynamics simulation. We thank Haiyan Chen from

Qichun Wei's lab for providing the luciferase transfected B16F10. We thank Qin Han, Chenyu Yang, Dandan Song and Guizhen Zhu at the Center of Cryo-Electron Microscopy (CCEM), Zhejiang University for their technical assistance on confocal laser scanning microscopy, TEM and SEM. We thank Yanwei Li at the Core Facilities of Zhejiang University School of Medicine for technical assistance in flow cytometry analysis. We thank Zhiwei Ge and Xiaodan Wu at Analysis Center of Agrobiology and Environmental Sciences, Zhejiang University for technical assistance in tandem mass spectrometry and liquid chromatography-tandem mass spectrometry.

## Author contributions

M.H., Y.Z. and J.G. conceived the project and designed the experiments. M.H. and J.G. supervised the project, discussed and commented on the manuscript. Y.Z. performed the majority of the experiments and data analysis. P.K. and X.B. assisted with cell culture of B16F10, preparing MCM and animal experiments. H.W. assisted with the establishment of animal models. Y.X. synthesized TPE-FFKY. Y.Z. performed the flow cytometry experiments with assistance from Z.Z., T.W., M.L., P.K., Y.S.L., Y.X., H.Z., X.Z. and Q.Y. Y.Y.L. supported the operation of confocal microscope. Y.Z., Q.D., P.K., H.Z., Y.X., Y.S.L., X.J. and H.W. operated the tumor resection surgery. Y.Z. wrote the manuscript. L.W. polished the article. All authors discussed the results and commented on the manuscript.

## Competing interests

The authors declare no competing interests.
