## [Peer Review File · Nature Communications]

REVIEWER COMMENTS

Reviewer #1 (Remarks to the Author):

This manuscript reported that an MMP-2 activatable peptide assembly FR17 impedes the establishment of primary tumor induced pre-metastatic niche (PMN) as a 'flame-retarding blanket'. The peptide FR17 consists of two main parts, which is (1) the self-assembly peptide domain FFKY and (2) thymopentin with good hydrophilic property and immune modulation effect. Considerable amount of in vitro experiments were done to illustrate the successful inhibition of PMN formation from different aspects, including the interruption of the activation of fibroblast, the inhibition of the vascular leakage and angiogenesis, the impediment of MDSC recruitment and modulatory of the immune microenvironment. Besides, In vivo experiments were further verified the FR17 treatment inhibits melanoma lung metastasis. Although the MMP2 responsive behavior of peptides in vivo remains to be confirmed, and low efficiency of MMP2 implied in the limitation of the designed peptide, this work, indeed, illustrates the promises of using enzymatic self-assembly for treating cancer. Thus, I support the acceptance of this work if the authors were able to addressing the following issues.

1. The key problem is the lack of the validation of the nanoblankets. The morphology changes before and after enzyme treatment should be compared to verify the in-situ formation of this blanket like structure. This experiment is imperative to support the claim of nanoblankets in vivo.
2. MS data of peptide sequence after MMP2 treatment should be given to verify the enzymatic response of designed peptides.
3. The concentrations of peptides used for TEM measurement can be listed in the caption of Figure 1.
4. There are some outstanding works about the enzymatic responsive peptides to induce cancer cell death or inhibit tumor growth, for example 10.1021/ja510156v and j.chempr.2019.06.020. The authors should properly refer the closely related works.
5. FFKY is not a part of b-amyloid since the sequence of A-beta is "DAEFRHDSGYEVHHQKLVFFAEDVGSNKGAIIGLMVGGVVIA". Refs14,15 are not the first use of FFKY in self-assembly. The author should give a more proper reference of the FFKY in the literature.

Reviewer #2 (Remarks to the Author):

This article applied peptide nano-blanket to impede pre-metastatic niche formation in the lung, especially by suppressing fibroblast activation.

The article is interesting, revealing great potential for newly developed technologies, and fascinating, showing possible treatment prior to cancer cell arrival for decreasing metastasis. This reviewer does agree that this story is important and is novel, however, this reviewer feels that there are several issues that need to be addressed:

- 1) My biggest concern is that authors didn't show where and how nano-blanket is attached. The authors need to show in some way how much nano-blanket resides within the lung and also in other organs. They claim that this is site specific, but liver pre-metastatic niche formation should also be occurring. Is it

really lung specific?

2) In the same note as above, how long does nano-blanket stay in the organ. Based on figure 6, it seems like metastasis in the lung is just simply delayed. Would it be possible to keep the treatment going and would that truly impede metastasis.

3) Also, it would be important to show whether nano-blanket is degradable. When would it degrade after landing to the lung. Or would it stay without degrading.

4) With the current treatment, does authors believe that they are flaming out all activated fibroblasts in the lung, or is it partial flame-out. It was not clear how they decided upon the injection amount.

5) In figure 4, authors show that activated fibroblast CM induces angiogenesis and nano-blanket treated CM suppresses this effect. It is not clear whether there are any left-over nano-blanket in the CM, which could directly impede angiogenesis. Authors need to show that there are no nano-blanket left in the CM.

6) Minor point: Please show in figure 2a an arrow pointing to what we should be looking at. It is hard to appreciate what we should be seeing. Also, FFKY image looks completely different from the others. Was this expected?

7) Minor point: In figure 3f, invasion assay and contraction assay should be labeled separately.

8) Minor point: In figure 4d, MCM free control also shows big gap in the current image. Quantitative analysis is needed and an image that represents the quantitative data needs to be replaced.

9) Minor point: Although bone-marrow recruited cells, especially MDSCs, are analyzed in the data, not much is mentioned in the introduction. It would be helpful for the readers to include more information on why this analysis is necessary.

Reviewer #3 (Remarks to the Author):

In this manuscript, the authors constructed an enzyme-activatable peptide FR17 to release self-assembly monomer FG8 to form nano-blanket in PMN microenvironment for impeding fibroblasts activation and preventing metastatic cascades. The design is interesting, and the authors did comprehensive work to justify their hypothesis. The presentation of the work is also clear and neat. The following comments can be considered for improvement.

1. In page 3 line 14, the authors stated that nano-blanket in PMN can impede fibroblasts activation to prevent metastatic. Why did the authors choose fibroblasts as the target of intervention? As primary tumor-derived cytokines and exosomes and myeloid-derived suppressor cells (MDSCs) all contribute to metastasis, it is necessary to provide the background about the keys factors influencing PMN microenvironment, and discuss the role of fibroblasts in PMN and the current studies made in this field based on fibroblasts.

2. Is there any direct evidence about the production of FG8 in the presence of MMP2?

3. In supplementary Figure 4, FG8 was shown to achieve self-assembly in water? Did the authors investigate self-assembly of FG8 under different pH conditions? The authors should consider the condition in PMN and investigate the self-assembly of FG8 under the corresponding condition.

4. Please provide semi-quantitative data in Figure 3g, 4e, 5c, supplementary Figure 11 and 12.

5. Why did the authors choose subcutaneous injection to administrate FR17 instead of intravenous injection? Please explain.

** See Nature Research's author and referees' website at www.nature.com/authors for information about policies, services and author benefits.

Point-by-point Response to the Reviewers' Comments

Reviewer #1 (Remarks to the Author):

This manuscript reported that an MMP-2 activatable peptide assembly FR17 impedes the establishment of primary tumor induced pre-metastatic niche (PMN) as a 'flame-retarding blanket'. The peptide FR17 consists of two main parts, which is (1) the self-assembly peptide domain FFKY and (2) thymopentin with good hydrophilic property and immune modulation effect. Considerable amount of *in vitro* experiments were done to illustrate the successful inhibition of PMN formation from different aspects, including the interruption of the activation of fibroblast, the inhibition of the vascular leakage and angiogenesis, the impediment of MDSC recruitment and modulatory of the immune microenvironment. Besides, *in vivo* experiments were further verified the FR17 treatment inhibits melanoma lung metastasis. Although the MMP2 responsive behavior of peptides *in vivo* remains to be confirmed, and low efficiency of MMP2 implied in the limitation of the designed peptide, this work, indeed, illustrates the promises of using enzymatic self-assembly for treating cancer. Thus, I support the acceptance of this work if the authors were able to addressing the following issues.

1. The key problem is the lack of the validation of the nanoblankets. The morphology changes before and after enzyme treatment should be compared to verify the in-situ formation of this blanket like structure. This experiment is imperative to support the claim of nanoblankets *in vivo*.

Response: We appreciate your comments and suggestions and totally agree. The morphology changes before and after enzyme treatment of FR17 has been observed *via* TEM. With hydrophilic property, FR17 wouldn't aggregate in aqueous solution, as indicated in Figure 1a below, which couldn't be measured and analyzed by the particle size meter (Zetasizer Nano ZS, Malvern Instruments), while the peptide self-assemblies aggregated by enzymatic degradation with an average diameter of 500~800 nm was showcased in Figure 2b in the manuscript. The lamellar structure of the peptide nano-blanket was pseudo-colored in gold in TEM and Cryo-TEM images (Figure 1a, b). When added enzyme to FR17 and left to stand for 72 h, a visible semitransparent thin layer is woven in the aqueous system, which can be easily ripped up by gently shaking and split into smaller pieces through sharply shaking (Figure 1c). Besides, the specific microenvironment-responsive assembly of peptide nano-blanket was validated at the cellular level on fibroblasts. As indicated by Figure 1d, peptide nano-blanket (pseudo-colored in gold) was constructed on the cell surface of fibroblasts in the condition of melanoma-conditioned media (MCM) cultivation, for MCM contains and also increases the

secretion of MMPs.

Figure I. Peptide nano-blanket transformed from FR17. **a**, TEM images of FR17 (500 μ M) before and after MMP2 treatment (1 μ g/ml). Scale bar = 200 nm. **b**, Cryo-TEM image of the peptide nano-blanket assembled by FR17 (500 μ M) treated with MMP2 (1 μ g/ml). Scale bar = 200 nm. **c**, Macroscopic images of the thin layer formed by FR17 treated with enzyme and let stand for 72 h. The soft thin layer broke into pieces after gently shaking and dispersed into nanoscale fragment after sharply shaking. **d**, SEM images of the peptide nano-blanket assembled on the cell surface of fibroblasts in the condition of MCM inducement. The in-situ formed peptide nano-blanket on the surface of fibroblasts is pseudo-colored in gold. Scale bar = 1 μ m in the macro images. Scale bar = 500 nm in the enlarged images circled with dash line.

What's more, in order to demonstrate the *in-situ* assembly of the peptide nano-blanket *in vivo*, TPE-FR17 was administrated to PMN mice. It has been demonstrated in Supplementary Figure 6 in the manuscript that FR17 modified with the aggregation-induced emission (AIE) luminogens TPE endows peptide with the fluorescence "turn-on" effect when TPE-FR17 was cleaved to release TPE-FG8 which would spontaneously aggregate. In short, TPE-FR17 was subcutaneously administrated to PMN mice (on Day 10) and healthy mice at the dose of 100 $\mu\text{M}/\text{kg}$. 12 h-post peptide administration, mice were euthanized for cardiac perfusion and the lung tissues were collected for frozen sections. The frozen sections were observed under high-resolution laser confocal microscopy excited by 405 nm laser. As indicated in Supplementary Figure 7 as follow, fluorescent dots were observed in PMN lung with higher expression of MMP2 (Supplementary Figure 8d) rather than in healthy lung, indicating the specific enzymatic cleavage of TPE-FR17 and the *in-situ* assembly of the peptide nano-blanket in PMN lung.

Supplementary Figure 7. The aggregation-induced emission effect of the in-situ assembly of peptide nano-blanket at 12 h-post subcutaneous administration of TPE-FR17 in healthy or PMN lung. Peptide assemblies of the monomer TPE-FG8 released from TPE-FR17 in PMN were pseudo-colored in white for contrast on the upper panel and pseudo-colored in red in the merged images. Scale bar = 200 μm .

2. MS data of peptide sequence after MMP2 treatment should be given to verify the enzymatic response of designed peptides.

Response: Thank you for your suggestion, which helps to improve and perfect our manuscript. Though we had considered the MMP2-cleavable peptide linker PLGLAG as the confirmed and well-applied MMP2 recognition sequence, it's essential to verify the enzymatic response of the designed peptides in this work as well. Therefore, LC-MS/MS was applied to verify the production of FG8 from FR17 or sFD17 after MMP2 treatment. Briefly, peptide FR17 or sFD17

(200 μM) was cultivated with pre-activated hMMP2 (200 ng/ml) for 10 min. The reaction was terminated by adding 3 volume of methanol and centrifuged at 1,3000 rpm for 15 min to discard the protein precipitation. Samples were analyzed by LC-MS/MS. Reference FG8 (the MS and MS/MS of FG8 has been added in Supplementary Figure 1 & 2 in the revised manuscript), and FR17 or sFD17 without enzyme treatment were analyzed as standard control. As demonstrated by Supplementary Figure 3 as follow, the retention time of FG8 ($t_R = 4.9$ min) is ahead of FR17 ($t_R = 5.5$ min) or sFD17 ($t_R = 5.3$ min) in chromatogram. After MMP2 cultivation for 10 min, the characteristic peak of FG8 aroused. What's more, the peptide sequence produced by MMP2 degradation was able to be identified under the auto-optimized condition of the ion pair 464.8/379.6, in which 464.8 represents $[\text{M}+2\text{H}]^{2+}$ of FG8 (Exact mass 927.49) and 379.6 represents $[\text{M}+2\text{H}]^{2+}$ of FFK(GP)Y (Exact mass 757.38). By calculation, the responsive rate of MMP2 to FR17 is 34.28% in 10 min, the responsive rate of MMP2 to sFD17 is 35.82% in 10 min.

Supplementary Figure 3. Enzyme cleavage of FR17 and sFD17 to release the self-assembled monomer FG8. **a**, Schematic of the enzyme cleavage of FR17 or sFD17. **b**, Liquid chromatography-tandem mass spectrometry (LC-MS/MS) of FR17 and sFD17 (200 μM) before and after MMP2 (200 ng/ml) treatment. The characteristic peak of FG8 was acquired under the auto-optimized condition of the ion pair 464.8/379.6, in which 464.8 represents $[\text{M}+2\text{H}]^{2+}$ of FG8 (Exact mass 927.49) and 379.6 represents $[\text{M}+2\text{H}]^{2+}$ of FFK(GP)Y (Exact mass 757.38).

3. The concentrations of peptides used for TEM measurement can be listed in the caption of Figure 1.

Response: Thank you for your reminder. The concentrations of peptides have been listed in the figure caption in the revised manuscript.

Figure 2. Enzyme-activated self-assembly of FR17 and all-atom molecular dynamics (MD) simulation of the self-assembly of FG8. a, TEM images of FR17 or sFD17 (500 μ M) treated with MMP2 (1 μ g/ml) (scale bar = 200 nm), FG8 (500 μ M) and FFKY (500 μ M) (scale bar = 100 nm). b, Size distribution of FR17 and sFD17 (500 μ M) after MMP2 (1 μ g/ml) cleavage.

4. There are some outstanding works about the enzymatic responsive peptides to induce cancer cell death or inhibit tumor growth, for example 10.1021/ja510156v and j.chempr.2019.06.020. The authors should properly refer the closely related works.

Response: Thank you very much for your suggestions. Some great works which are closely related to our research have been cited in the introduction as follow, including 10.1021/ja510156v and j.chempr.2019.06.020 as you suggested.

“Inspired by the self-assembled peptides found in many natural life processes, researchers have modified, designed and synthesized diverse self-assembled peptides applied as functional biomaterials for various applications^{1, 2, 3}. In the application on anti-tumor and anti-metastasis therapy, enzyme-responsive self-assembly or ligand-receptor interactions-triggered morphology transforming of the peptide nanofibrils and hydrogel have been developed to induce cell death^{4, 5, 6}, to restrict tumor cell invasion⁷, to serve as biocompatible drug delivery platforms⁸, to achieve specific targeting of tumor⁹ and imaging¹⁰ as well.”

5. FFKY is not a part of b-amyloid since the sequence of A-beta is “DAEFRHDSGYEVHHQKLVFFAEDVGSNKGAIIGLMVGGVVIA”. Refs14,15 are not the first use of FFKY in self-assembly. The author should give a more proper reference of the FFKY in the

literature.

Response: Thank you for your suggestion. The reason why the former sentence was organized as “...the backbone of a self-assembly peptide domain Phe-Phe-Lys-Tyr (FFKY), which is derived from β -amyloid (A β) peptide.” is to trace back to the fountainhead of FFKY, dipeptide FF, which is extracted from the main fragment of A β with self-assembly property¹¹. To avoid misreading, this sentence has been revised as follow:

“...the backbone of a self-assembly peptide domain Phe-Phe-Lys-Tyr (FFKY), a variant of Phe-Phe (FF), which is derived from the Alzheimer's β -amyloid (A β).”

In addition, as the original fragment of A β ₁₆₋₂₀, KLVFF has been applied as therapeutic agents^{7, 12}, imaging agents¹³ or delivery platform¹⁴ in the field of Alzheimer's disease¹⁵ and cancer¹⁶. Further researches cut the peptide fragment down to dipeptide FF to obtain discrete nanotubes through self-assembly in aqueous solution¹. Afterwards, felicitous modification and re-designment on FF with perfect biocompatibility has been reported with a wide range of applications¹⁷. For example, FFKF was developed to construct drug delivery system which can be completely degraded by cathepsin proteases¹⁸. Yang and his team employed FFFK to develop molecular hydrogel for co-delivery of anti-cancer drugs¹⁹. In another work, the application of FFYK was explored in organelles targeting and cancer cell killing²⁰. Some introduced naphthyl group on N terminal of Phe to favor intercellular hydrophobic interactions²¹.

Here in our manuscript, taking advantages of both the self-assembly feature of diphenylalanine structural motif FF with the assistance of Y and the editable site provided by K to combine with hydrophilic fragment *via* enzyme-cleavable linker, FFKY was employed as the backbone of the self-assembled monomer FG8, which would spontaneously fold into lamellar structure, constructing the peptide nano-blanket.

Reviewer #2 (Remarks to the Author):

This article applied peptide nano-blanket to impede pre-metastatic niche formation in the lung, especially by suppressing fibroblast activation.

The article is interesting, revealing great potential for newly developed technologies, and fascinating, showing possible treatment prior to cancer cell arrival for decreasing metastasis. This reviewer does agree that this story is important and is novel, however, this reviewer feels that there are several issues that need to be addressed:

1) My biggest concern is that authors didn't show where and how nano-blanket is attached. The authors need to show in some way how much nano-blanket resides within the lung and also in other organs. They claim that this is site specific, but liver pre-metastatic niche formation should also be occurring. Is it really lung specific?

Response: Thank you for your suggestions. With the MMP2-cleavable PLGLAG peptide linkage, FR17 can be degraded to release self-assembly monomer FG8 at the site highly-expressed with MMP2, which is secreted by the stromal cells to the intercellular substance or localize to the cell surface by binding to integrins^{22, 23}. Therefore, the dissociated FG8 aggregate to construct peptide nano-blanket on the cell surface or in the intercellular substance *in-situ*. As directly viewed by SEM of the fibroblasts induced by MCM with peptide treatment and the STEM images of the PMN lung collected from the model mice received peptide administration, Figure II exhibits the construction of nano-blanket on cell surface under certain conditions *in vitro* (Figure IIa) and its suspected appearance in the intercellular substance in PMN lung *in vivo* (Figure IIb), and the morphology and the size of which ranges from hundreds to thousands nanometers depending on the intercellular space specifically at PMN sites.

Figure II. The *in-situ* assembled peptide nano-blanket *in vitro* and *in vivo*. **a**, SEM images of the peptide nano-blanket assembled on the cell surface of fibroblasts in the condition of MCM inducement. The *in-situ* formed peptide nano-blanket on the surface of fibroblasts is pseudo-colored in gold. Scale bar = 1 μm in the macro images. Scale bar = 500 nm in the enlarged images circled with dash line. **b**, STEM images of the peptide nano-blanket in the intercellular substance in PMN lung, which was collected from PMN mouse at 12 h-post subcutaneous administration of FR17. Scale bar = 1 μm .

To reveal the *in-situ* assembly and distribution of the peptide nano-blanket *in vivo*, TPE-FR17 was administrated to PMN mice. It has been demonstrated in Supplementary Figure 6 in the manuscript that FR17 modified with the aggregation-induced emission (AIE) luminogens TPE endows peptide with the fluorescence “turn-on” effect when TPE-FR17 was cleaved to release TPE-FG8 which would spontaneously aggregate. In short, TPE-FR17 was subcutaneously administrated to PMN mice (on Day 10) and healthy mice at the dose of 100 $\mu\text{M}/\text{kg}$. At the time point of 1, 2, 4, 8, 12, 24, 48 h-post peptide administration, mice were euthanized for cardiac perfusion and the lung tissues were collected for frozen sections. The frozen sections were observed under high-resolution laser confocal microscopy excited by 405 nm laser. As indicated in Figure III as follow, fluorescent dots were observed in PMN lung with higher expression of MMP2 (Figure IV) rather than in healthy lung, indicating the specific enzymatic cleavage of TPE-FR17 and the *in-situ* assembly of the peptide nano-

blanket in PMN lung. What's more, according to the density of the fluorescent dots presented in the lung sections, the peptide nano-blanket assembled in 1 h after peptide administration, and the process of construction and increased accumulation of the peptide nano-blanket last for 12 h, which might be contributed by the sustained release of TPE-FR17 from the subcutaneous drug storage naturally formed by subcutaneous injection.

Figure III. The AIE effect of the in-situ assembly of peptide nano-blanket from 1 h to 48 h-post subcutaneous administration of TPE-FR17 in PMN lung. The lung collected at 12 h-post subcutaneous administration of TPE-FR17 to Healthy mouse was set as control.

Peptide assemblies of the monomer TPE-FG8 released from TPE-FR17 in PMN were pseudo-colored in white for contrast on the upper panel and pseudo-colored in red in the merged images. Scale bar = 200 μm . The semi-quantification was calculated from 6 visual fields per time-point *via* ImageJ.

We do acknowledge the wide-reported formation of liver PMN in diverse tumors^{24, 25, 26}. However, in the PMN model we've referenced²⁷, both the pioneers and our team focused on the pulmonary pre-metastatic and metastatic niches. Though the MCM inducement (which contains tumor-derived secreted factors) might arise broad alterations on the cellular and molecular level as well as the local microenvironment changes, especially in some certain target organs including lung, liver, etc.^{28, 29} In this model, B16F10 melanoma cells were intravenously administrated as the simulation of circulative tumor cells in the spontaneous process in other metastatic models, who would wander through blood vessels and some of which would successfully colonize in the prepared PMN in lung. Given the circumstances of how this PMN model was established and the fact that no hepatic metastases were observed during the entire pathological process on our PMN model mice (Supplementary Figure 21), our research focused on the pulmonary pre-metastatic niche and the further developed pulmonary metastases.

As we demonstrated, the *in-situ* assembly of peptide nano-blanket is enzyme-activated at the site with higher expression of MMP2, which is considered as a site-specific hall-marker of PMN^{30, 31}. Therefore, the site-specific assembly of the peptide nano-blanket might not be lung specific in the patients with multiple pre-metastatic sites. To determine whether the responsive assembly of the peptide nano-blanket would occur in liver in the PMN model adopted in the research, MMP2 expression in lung and in liver tissues collected from healthy or PMN mice has been compared. As suggested by Figure IV, the highest expression of MMP2 was detected in PMN lung. Meanwhile, there's a slight enhancement of MMP2 expression in PMN liver when compared to healthy liver, which remains in the lower level than that of PMN lung.

Figure IV. MMP2 expression in healthy or PMN lung and liver on Day 10. H represents for healthy. PMN model was established as described in Method, ie. mice were intraperitoneally injected with melanoma-conditioned medium (300 μl per mice) for 10 consecutive days, and a tail vein injection of B16F10 cells (1×10^5 per mice) was given to the mice on Day 7.

Furthermore, liver tissues as well as other major organs including heart, spleen and kidney were also collected for frozen sections at the time point of 1, 2, 4, 8, 12, 24, 48 h-post peptide administration. The frozen sections were looked over under high-resolution laser confocal microscopy excited by 405 nm laser to examine the AIE effect of peptide assembly. As showcased in Figure V, several fluorescent dots were observed in PMN liver sections with the same accumulation trend as it did in PMN lung in time dimension but with lesser peptide aggregates. For comparison, images of lung and liver sections at 12-h post peptide administration were arranged together in Figure VI. And no fluorescent dot of peptide aggregation has been observed in other organ sections as presented in Figure VII.

Figure V. The AIE effect of the in-situ assembly of peptide nano-blanket from 1 h to 48 h-post subcutaneous administration of TPE-FR17 in PMN liver. The liver collected at 12 h-post subcutaneous administration of TPE-FR17 to Healthy mouse was set as control.

Peptide assemblies of the monomer TPE-FG8 released from TPE-FR17 in PMN were pseudo-colored in white for contrast on the upper panel and pseudo-colored in red in the merged images. Scale bar = 200 μm . The semi-quantification was calculated from 6 visual fields per time-point *via* ImageJ.

Figure VI. The AIE effect of the in-situ assembly of peptide nano-blanket at 12 h-post subcutaneous administration of TPE-FR17 in PMN lung and liver. Peptide assemblies in PMN were pseudo-colored in white for contrast on the upper panel and pseudo-colored in red in the merged images. Scale bar = 200 μm .

Figure VII. Representative images of the heart, spleen and kidney after TPE-FR17 administration. Scale bar = 200 μ m.

2) In the same note as above, how long does nano-blanket stay in the organ. Based on figure 6, it seems like metastasis in the lung is just simply delayed. Would it be possible to keep the treatment going and would that truly impede metastasis?

Response: According to the in-situ peptide assembly reflected by AIE effect as illustrated in Figure III & V above, the peptide nano-blanket formed and further accumulated for 12 hours after subcutaneous administration of the peptide. Almost no fluorescent signal was observed on all organ sections at 24 h-post peptide administration, suggesting that the nano-blanket would constantly construct and stay for hours but would soon be degraded within 24 h.

The peptide FR17 was designed to response to construct peptide nano-blanket in PMN stromal microenvironment, exerting inhibition effect on fibroblasts activation so as to prevent metastatic cascades. In this work, we would like to more concentrate on the process of PMN development, emphasizing on the early intervention on metastasis of FR17. Therefore, we explored further on the subsequent impact of inhibition on fibroblasts activation induced by FR17 intervene, revealing

the underlining mechanism on cellular interactions among fibroblasts, vascular endothelial cells and extracellular components, and intervention on PMN recruited MDSCs. As for whether it's possible to achieve complete suppression on metastasis by tailoring the treatment plan, a preliminary agreement has been made after group discussion. As commonly acknowledged as a complex multistep process, tumor metastasis is promoted and led by multiple factors, including not only the preparation of PMN but also other vital contributors such as the specific phenotypes and invasion ability of tumor cells, the developing stages at the exact time point when take measures for drug intervention or surgical operation. In the animal models we've established for the investigation on PMN, a half million highly invasive malignant cells in rapid proliferation were administrated in one single injection to the mice directly through the blood circulation. This simulation of the tens and thousands of circulative tumor cells wondering in the circulation would be diagnosed as the terminal stage of tumor with high aggressiveness in clinic and would be fatal to the patients. Though FR17 administration has been verified to serve as a "flame-retarding blanket" at PMN site specifically to extinguish the "fire" of tumor-supportive microenvironment adaption, FR17 exhibits only suppression on tumor cells migration but no significant anti-tumor effect (Supplementary Figure 19) because the peptide carrying no anti-tumor therapeutics at all. Accordingly, to completely remove all of the metastatic lesions in rapid growth, the co-administration of FR17 and other anti-tumor therapeutics with tumor-killing effect might work, which would be appealing to us in our further researches in the future.

3) Also, it would be important to show whether nano-blanket is degradable. When would it degrade after landing to the lung? Or would it stay without degrading.

Response: Yes, most of the peptide assemblies were biodegradable *in vivo* with high biocompatibility^{18, 32}, which is one of their superiorities valued in biomedical applications. As illustrated in Figure III & V above, the peptide nano-blanket was constructed and further accumulated for 12 h after subcutaneous administration of the peptide, all of which would be degraded within 24 h with no fluorescent signal being observed on all organ sections at 24 h-post peptide administration. Furthermore, the preliminary safety evaluation of peptide administration on lung metastasis model with MCM-induced PMN suggested the bio-safety of the peptide nano-blanket. Long-term administration of FR17 or sFD17 has no significant hepatic toxicity on PMN model mice (Supplementary Figure 20b). No organic lesions in H & E organ sections was observed (Supplementary Figure 21).

4) With the current treatment, does authors believe that they are flaming out all activated fibroblasts in the lung, or is it partial flame-out. It was not clear how they decided upon the injection amount.

Response: After the administration of FR17 or sFD17 for one week, most of the fibroblasts presented Vimentin⁺αSMA⁻ resting state in the lung of PMN model mice as illustrated in the immunofluorescence-stained lung sections in Figure VIII below. Yet it's inaccurate to draw the conclusion that all of the Vimentin⁺αSMA⁺ activated fibroblasts in the lung have been beaten out. Different subsets of fibroblasts, including resting and activated fibroblasts, are both considered to be essential to maintain lung architecture and function, playing a role in regulating air and blood relationships by modulating alveolar geometry. It's reported that there's a small population of activated fibroblasts resident even in the healthy lung. The conversion between its resting and activated phenotype maintains a delicate balance. Back to 1970s, it has been suggested that activated lung fibroblasts contribute to immune defense and protect the lung against proteolytic damage. Yet in PMN development, the over production of extracellular matrix and chemotactic factors by the over-aroused fibroblasts go out-of-balance, recruiting immune suppressor cells, heading to the direction of pro-metastasis lesion. Therefore, the wise way is to contain the over-rousement of activated fibroblasts in advance, which has been indicated by the data provided as follow (Supplementary Figure VIII) and in the manuscript from several aspects of both cell markers and bio-functions, production of extracellular matrix (ECM), ECM-remodeling enzymes and cytokines (Figure 3, Supplementary Figure 13).

Figure VIII. Representative images of Vimentin⁺αSMA⁺ activated fibroblasts in the lung harvested from mice administrated with different peptides on Day 10. Scale bar = 50 μm.

The drug dose was determined according to former applications and the preliminary experiments. Referred studies on both peptide assemblies^{6, 7, 33} and the clinical evaluation of hydrophilic drug TP5^{34, 35}, mice in different groups were treated with a low (20 μM/kg), medium (40 μM/kg) or high (80 μM/kg) dose of peptide in the preliminary experiment. The therapeutic outcome of

medium dose is close to the high dose groups, which is better than that of low dose group (Data not shown). On the other hand, according to the determined enzyme-responsive assembly of FR17 (200 μM in response to 200 ng/ml MMP2), the suitable dose of FR17 administration was calculated and further applied as 40 $\mu\text{M}/\text{kg}$, with the peptide concentration reaching to higher than 500 μM *in vivo*. Though the current data illustrated the success in flaming out fibroblast activation induced by tumor-derived secretion factors both *in vitro* (Figure 3a-i, Supplementary Figure 12) and *in vivo* (Figure 3j-k, Supplementary Figure 13, Figure VIII). Still, further exploration of the possible dose-dependent manner of the peptide nano-blanket or FR17 on the suppression extent of pre-metastasis associated fibroblast and the mechanism underneath is required for its clinical applications in the future. Thank you very much for your question.

5) In figure 4, authors show that activated fibroblast CM induces angiogenesis and nano-blanket treated CM suppresses this effect. It is not clear whether there are any left-over nano-blanket in the CM, which could directly impede angiogenesis. Authors need to show that there are no nano-blanket left in the CM.

Response: We apologize for the unclear statement in the method that led misunderstanding. There was no nano-blanket nor peptide being left to directly influence angiogenesis. The description of the experimental procedure on this section has been revised in detail to help with comprehension. As illustrated in Supplementary Figure 14a below, MLF-conditioned medium (FCM) was obtained as follows: MLFs were stimulated by MCM (supplemented with 10% (v/v) FBS) with or without peptide drugs (TP5, sFD17, FR17 at the concentration of 100 μM) for 48 h, the original medium was discard, then replaced with fresh complete medium after PBS rinsing to eliminate the direct influence of MCM and peptide drugs on the later experiment subject. After incubated for another 24 h, the cell supernatants were collected, centrifuged at 2000 rpm for 10 min to discard the cell debris. Thus, the replacement of the culture medium ensured to get rid of the direct effect of MCM and peptide drugs on the endothelial cells.

Supplementary Figure 14a, The experimental procedure to obtain the conditional FCM after MCM stimulation and the peptide treatment on MLF *in vitro* for further experiments on endothelial cells.

6) Minor point: Please show in figure 2a an arrow pointing to what we should be looking at. It is hard to appreciate what we should be seeing. Also, FFKY image looks completely different from the others. Was this expected?

Response: Sorry for the misreading caused by the presented TEM images. We've pseudo-colored the lamellar structure of the peptide nano-blanket in gold in the following images, hoping this would help to make out these overlapped curly thin layers. In addition, it's exactly right that the morphology of FFKY assemblies is completely different from the peptide nano-blanket, which is assembled by FG8. With the high hydrophobicity offered by multi-aromatic groups, FFKY would assemble into nanofibers/nanotubes *via* intermolecular π - π effects, which could be similar to the morphology of FFKF assemblies as formerly reported¹⁹.

Figure 2a. TEM images of FR17 or sFD17 (500 μM) treated with MMP2 (scale bar = 200 nm), FG8 (500 μM) and FFKY (500 μM) (scale bar = 100 nm).

7) Minor point: In figure 3f, invasion assay and contraction assay should be labeled separately.

Response: Thank you very much for your thoughtful suggestion. We've labeled two assays separately in Figure 3 in the revised manuscript.

Figure 3. FR17 interrupted the activation of fibroblast induced by tumor derived factors. **a.** Schematic drawing illustrates the *in-situ* assembled peptide nano-blank interrupts the activation of fibroblast. When activated by tumor derived factors during PMN development, the expression of proangiogenic factors and ECM remodeling factors would be up-regulated in fibroblast, as well as the ECM components production. While the peptide nano-blank could calm down fibroblast activation, down-regulating the above factors. **b.** Schematic illustration to show the protocol of MCM stimulation and peptide treatment on mice lung fibroblast (MLF) *in vitro*. **c.** Cell proliferation of MLF after MCM stimulation and peptide treatment. $n = 4$. **d.** Secretion of MMP9 ($n=4$), VEGF ($n = 4$ for peptide treated groups and 3 for the control groups) and fibronectin ($n=4$)

in the culture media of MLF after stimulated by MCM, treated with or without peptide. **e**, qPCR analysis of Acta2 (i.e., α Sma), Mmp9, Vegfa, Fibronectin1 (Fn1) expression in MLF after MCM stimulation and peptide treatment. **f & g**, Migration assay to evaluate MLF migration ability after MCM stimulation and peptide treatment. n = 4. Scale bar = 100 μ m. **h & i**, Collagen gel contract assay to evaluate contracting function of MLF after MCM stimulation and peptide treatment. n = 3. **j**, Expression level of fibronectin in the lung harvested from the PMN model mice administrated with different peptides on Day 10. **k**, Representative images and semi-quantification of α SMA⁺ fibroblasts in the lung harvested from mice administrated with different peptides on Day 10. Data is presented as mean \pm SD. n = 6. One-way ANOVA followed by Tukey's multiple comparisons test was employed for statistical evaluation. Scale bar = 50 μ m.

8) Minor point: In figure 4d, MCM free control also shows big gap in the current image. Quantitative analysis is needed and an image that represents the quantitative data needs to be replaced.

Response: As presented in Figure 4d and Figure IX with enhanced brightness, endothelial cells tightly attached to each other with the distribution of strong VE-cadherin signals along the contact membrane in MCM free control group. It is probable that the cell that sprawled out in the middle of the enlarged image might has been confused with a big gap with low brightness. On the contrary, VE-cadherin signals along the boundary of the intercellular spaces left out by neighboring endothelial cells were lost in MCM and TP5 groups. To reflect the re-distribution of VE-cadherin on cell membrane, fluorescent signals alongside the diagonal line (as indicated in semitransparent yellow oblique line on the lower panel) were analyzed *via* ImageJ. The no-signal zones labeled in light golden represent for the intercellular gaps contributed by the dismissal of VE-cadherin, indicating the breakage in the integrality of endothelial. Quantitatively, the disruption on endothelial cell-cell connection led to dye leakage in endothelial transwell permeability assay (Figure X). In short, MLFs pre-treated with MCM and peptides were seeded at the lower well. The endothelial cells were seeded on a 0.4 μ m Transwell insert above the top of the well until grown to confluence. Rhodamine B-dextran was added to the upper insert on the endothelial cell layer. After 1 h incubation, the translocation of Rhodamine B-dextran from the insert to the lower well leaking through the endothelial cell layer was measured to reveal the relative permeability of the cell layer.

Figure IX, Integrity of the endothelial cell monolayer after cultivated with conditional FCM collected from MCM and peptide pre-stimulated MLF, indicated by VE-cadherin on the membrane. a, The white box and the white arrows in the enlarged images indicate the tight junction between the endothelial cells. While the yellow box and yellow arrows indicate the disruption of cell-cell connection. Scale bar = 50 μ m. **b,** The distribution of VE-cadherin on cell membrane, fluorescent signals alongside the diagonal line (as indicated in semitransparent yellow oblique line) were analyzed *via* ImageJ.

Figure X, Schematic illustration of the transwell permeability assay. Permeability of the endothelial cell layer *in vitro* when co-cultured with MLF pre-treated with MCM and peptide. The relative permeability was normalized by dividing the fluorescence signals of the treatment groups by the control group. Data is presented as mean \pm SD. n = 3. One-way ANOVA followed by Tukey's multiple comparisons test was employed for data analysis.

9) Minor point: Although bone-marrow recruited cells, especially MDSCs, are analyzed in the data, not much is mentioned in the introduction. It would be helpful for the readers to include more information on why this analysis is necessary.

Response: Thank you for your suggestion. The introduction of how MDSCs take part in PMN and metastasis has been added as follow:

“In 2005, Lyden et al. brought to light the recruitment of VEGFR⁺ myeloid progenitor cells to PMN by localized FN deposition³⁶, which would be over-produced by activated resident fibroblasts. This specific cell population and its subtypes were then unified and classified as MDSCs with its potent capability to suppress immune responses³⁷, who make major contributions in developing immunosuppressive microenvironment *via* activation of nitric oxide (NO) signaling or reactive oxygen species (ROS) pathway³⁸.”

Besides, necessity of the researches on MDSC in our study has been brought up in the relevant sections in Page 16 as follow:

“Given the fact that MDSC occupies a vital position in PMN construction and metastasis formation, MDSC has attracted wide concerns in recent years...The recruitment of MDSC was found to be related to the enhancement of ECM production and remodeling in PMN, including cross-linking collagen by LOX secreted by tumor cells³⁹, FN deposition³³, periostin (POSTN)-enrichment⁴⁰. As MDSC plays a crucial role in developing immunosuppressive and inflammatory microenvironment, some pioneers have demonstrated that blocking the recruitment of MDSC could be a promising strategy to suppress early metastasis by preventing the development of breeding ground for tumor^{41, 42, 43, 44}.”

Reviewer #3 (Remarks to the Author):

In this manuscript, the authors constructed an enzyme-activatable peptide FR17 to release self-assembly monomer FG8 to form nano-blanket in PMN microenvironment for impeding fibroblasts activation and preventing metastatic cascades. The design is interesting, and the authors did comprehensive work to justify their hypothesis. The presentation of the work is also clear and neat. The following comments can be considered for improvement.

1. In page 3 line 14, the authors stated that nano-blanket in PMN can impede fibroblasts activation to prevent metastatic. Why did the authors choose fibroblasts as the target of intervention? As primary tumor-derived cytokines and exosomes and myeloid-derived suppressor cells (MDSCs) all contribute to metastasis, it is necessary to provide the background about the key factors influencing PMN microenvironment, and discuss the role of fibroblasts in PMN and the current studies made in this field based on fibroblasts.

Response: Thank you for your suggestion. As we mentioned in the introduction, there're several main participators contribute the construction of PMN, including 1) primary tumor-derived cytokines and exosomes, 2) myeloid-derived suppressor cells (MDSCs), and 3) the tumor re-educated stromal environment, which consists of pre-metastasis associated fibroblasts, destabilized vasculature and extracellular matrix (ECM). Though few researches have focused on PMN-associated fibroblasts, cancer-associated fibroblasts have received considerable attention in the past decades as accomplices of cancer cells to promote tumor development and to establish the metastatic niche^{45, 46, 47}. To organize the above defining factors in a clearer way, we've revised the introduction as follow:

“Relevant studies revealed that complex interactions between multiple participators and alteration in regulative pathways energized the construction of PMN, such as primary tumor-derived cytokines and exosomes, myeloid-derived suppressor cells (MDSCs), and the tumor re-educated stromal environment including pre-metastasis associated fibroblasts, destabilized vasculature and extracellular matrix (ECM)^{12, 13} (Fig. 1a). In 2005, Lyden et al. brought to light the recruitment of VEGFR⁺ myeloid progenitor cells to PMN by localized FN deposition³³, which would be over-produced by activated resident fibroblasts. This specific cell population and its subtypes were then unified and classified as MDSCs with its potent capability to suppress immune responses⁴⁸, who make major contributions in developing immunosuppressive microenvironment *via* activation of nitric oxide (NO) signaling or reactive oxygen species (ROS) pathway⁴⁹. What's more, recently studies revealed the irritation of stromal cells especially fibroblasts in primary tumor and in distal site induced by tumor-derived secreted factors *via* STAT3 signaling³⁵ and JNK signaling³⁴

pathways. It's reported that tumor-educated fibroblasts serve to reconstruct ECM, induce angiogenic and pro-inflammatory response of endothelial cells, preparing a tumor supportive host stroma³³, which gives a clue to the tipping point of PMN initializing as fibroblast activation."

2. Is there any direct evidence about the production of FG8 in the presence of MMP2?

Response: Thank you for your advice. LC-MS/MS was applied to verify the production of FG8 from FR17 or sFD17 after MMP2 treatment. Briefly, peptide FR17 or sFD17 (200 μ M) was cultivated with pre-activated hMMP2 (200 ng/ml) for 10 min. The reaction was terminated by adding 3 volume of methanol and centrifuged at 1,3000 rpm for 15 min to discard the protein precipitation. Samples were analyzed by LC-MS/MS. Reference FG8 (the MS and MS/MS of FG8 has been added in Supplementary Figure 1 & 2 in the revised manuscript), and FR17 or sFD17 without enzyme treatment were analyzed as standard control. As demonstrated by Supplementary Figure 3 as follow, the retention time of FG8 (tR = 4.9 min) is ahead of FR17 (tR = 5.5 min) or sFD17 (tR = 5.3 min) in chromatogram. After MMP2 cultivation for 10 min, the characteristic peak of FG8 aroused. What's more, the peptide sequence produced by MMP2 degradation was able to be identified under the auto-optimized condition of the ion pair 464.8/379.6, in which 464.8 represents $[M+2H]^{2+}$ of FG8 (Exact mass 927.49) and 379.6 represents $[M+2H]^{2+}$ of FFK(GP)Y (Exact mass 757.38). By calculation, the responsive rate of MMP2 to FR17 is 34.28% in 10 min, the responsive rate of MMP2 to sFD17 is 35.82% in 10 min.

Supplementary Figure 3. Enzyme cleavage of FR17 and sFD17 to release the self-assembled monomer FG8. **a**, Schematic of the enzyme cleavage of FR17 or sFD17. **b**, Liquid chromatography-tandem mass spectrometry (LC-MS/MS) of FR17 and sFD17 (100 μ M) before and after MMP2 (200 ng/ml) treatment. The characteristic peak of FG8 was acquired under the auto-optimized condition of the ion pair 464.8/379.6, in which 464.8 represents $[M+2H]^{2+}$ of FG8 (Exact mass 927.49) and 379.6 represents $[M+2H]^{2+}$ of FFK(GP)Y (Exact mass 757.38).

3. In supplementary Figure 4, FG8 was shown to achieve self-assembly in water. Did the authors investigate self-assembly of FG8 under different pH conditions? The authors should consider the condition in PMN and investigate the self-assembly of FG8 under the corresponding condition.

Response: We appreciate you for arising such a good question. In the revision, we've compared the self-assembly property of FG8 in weak acidic condition to neutral pH condition. Since the biochemical condition in PMN has not been revealed or widely accepted, we've decided to adjust the pH value to 6.8 as a simulation condition of PMN *in vitro*, because a myriad of studies show that localized interstitial acidosis (pH 6.5–6.8) is a biochemical hallmark in inflammatory tissues^{50, 51}, ischemic organs⁵², and solid tumors⁵³. FG8 assembled in water, PBS (pH 7.4), PBS (pH 6.8) or

dissolved in methanol at the concentration of 100 μM gave similar performance on size distribution (100-200 nm, which is only for reference for the Zetasizer size meter is more applied to regular nanoparticles) and TEM morphology (the thin layer/membrane-like structure), indicating that FG8 is able to achieve assembly in both neutral and weak acidic conditions. Besides, TPE labeled FG8 with aggregation-induced emission effect was also employed to reflect the spontaneous aggregation of FG8 in different pH conditions. As illustrated in Supplementary Figure 5b, the aggregation of FG8 presented slightly stronger fluorescent signal in pH 6.8 than in neutral condition at the same concentration. Therefore, the following data provided evidence for the adaptability of the self-assembly of FG8 in different conditions.

Supplementary Figure 5. Self-assembly of FG8 in different pH conditions. **a**, Size distribution of the peptide assemblies of FG8 formed in water or PBS (pH 6.8 or pH 7.4). **b**, Fluorescence spectra of the TPE-FG8 (100 μM) in water, PBS (pH 6.8 or pH 7.4) or methanol (dissolved) excited by 405 nm. **c**, TEM images of FG8 assemblies in water, PBS (pH 6.8 or pH 7.4) or methanol (dissolved). Scale bar = 200 nm.

4. Please provide semi-quantitative data in Figure 3g, 4e, 5c, supplementary Figure 11 and 12.

Response: Thank you for your suggestion to improve the manuscript. The semi-quantitative data of Figure 3g, 4e, 5c, supplementary Figure 11 and 12 has been provided in the revised manuscript as follow.

Figure 3g (Labeled as **Figure 3j** in the revised manuscript)

Figure 4e

Figure 5c

Supplementary Figure 11 (Labeled as **Supplementary Figure 15** in the revised manuscript). **FR17 administration down-regulated matrix metalloproteinase in pulmonary PMN.** Representative images of MMP2 and MMP9 in the lung harvested from the PMN model mice administrated with different peptides. Semi-quantification was calculated from six random fields *via* ImageJ. Scale bar = 50 μm .

Supplementary Figure 12 (Labeled as **Supplementary Figure 16** in the revised manuscript). **FR17 administration prevented MDSC recruitment to pulmonary PMN by influencing the extracellular matrix remodeling.** **a**, Images under low magnification ratio of the serial sections of the lungs harvested from the model mice treated with different peptides were taken. Serial sections show the co-location and distribution of periostin (POSTN, red), lysyl oxidase (LOX, green) and Fibronectin (FN, red), CD11b⁺Gr1⁺ MDSC (yellow merged from green and red). Scale bar = 200 μm . **b**, The co-location analysis of CD11b⁺Gr1⁺ MDSC and POSTN, LOX, FN

alongside the yellow arrow marked on the above panel *via* ImageJ.

5. Why did the authors choose subcutaneous injection to administrate FR17 instead of intravenous injection? Please explain.

Response: As the most common administration route for peptides in clinic, subcutaneous injection is cost-effective and suitable for self-administration and repeated injections⁵⁴. More importantly, the short half-life in the bloodstream is the hallmark of peptide pharmacokinetics contributed by proteolytic cleavage by proteases and peptidases, renal clearance, liver metabolism and immunogenicity, especially for the peptides with high hydrophilicity⁵⁵. Compared to intravenous administration, drug could be sustained-released from the temporary formed drug storage under the skin after subcutaneous administration, maintaining serum concentrations and extending retention time^{56, 57, 58}. Moreover, the subcutaneous administration is bioequivalent in effect to intravenous injection, albeit with less between-patient variability⁵⁹. Overall, as a hydrophilic short chain peptide, FR17 was applied subcutaneously in this study to prolong its physiological effect *in vivo*.

References

- ¹ Ekiz MS, Cinar G, Khalily MA, Guler MO. Self-assembled peptide nanostructures for functional materials. *Nanotechnology* **27**, 402002 (2016).
- ² Abbas M, Zou Q, Li S, Yan X. Self-Assembled Peptide- and Protein-Based Nanomaterials for Antitumor Photodynamic and Photothermal Therapy. *Adv Mater* **29**, 1-16 (2017).
- ³ Loo Y, et al. Self-Assembled Proteins and Peptides as Scaffolds for Tissue Regeneration. *Adv Healthc Mater* **4**, 2557-86 (2015).
- ⁴ Tanaka A, et al. Cancer cell death induced by the intracellular self-assembly of an enzyme-responsive supramolecular gelator. *J Am Chem Soc* **137**, 770-5 (2015).
- ⁵ Feng Z, Han X, Wang H, Tang T, Xu B. Enzyme-Instructed Peptide Assemblies Selectively Inhibit Bone Tumors. *Chem* **5**, 2442-2449 (2019).
- ⁶ Guo, WW, et al. Intracellular restructured reduced glutathione-responsive peptide nanofibers for synergetic tumor chemotherapy. *Biomacromolecules* **21**, 444-453 (2020).
- ⁷ Hu XX, et al. Transformable Nanomaterials as an Artificial Extracellular Matrix for Inhibiting Tumor Invasion and Metastasis. *ACS Nano* **11**, 4086-4096 (2017).
- ⁸ Ben-Nun Y, et al. Cathepsin nanofiber substrates as potential agents for targeted drug delivery. *J Control Release* **257**, 60-67 (2017).
- ⁹ Chen W, et al. Combined Tumor Environment Triggered Self-Assembling Peptide Nanofibers and Inducible Multivalent Ligand Display for Cancer Cell Targeting with Enhanced Sensitivity and Specificity. *Small* **16**, e2002780 (2020).

-
- 10 Wang, TT, et al. AIE/FRET-based versatile PEG-Pep-TPE/DOX nanoparticles for cancer therapy and real-time drug release monitoring. *Biomater. Sci.* **8**, 118-124 (2020).
- 11 Reches M, Gazit E. Casting metal nanowires within discrete self-assembled peptide nanotubes. *Science* **300**, 625-7 (2003).
- 12 Tjernberg LO, et al. Arrest of beta-amyloid fibril formation by a pentapeptide ligand. *J Biol Chem* **271**, 8545-8 (1996).
- 13 Chandra Saha P, Das RS, Chatterjee T, Bhattacharyya M, Guha S. Supramolecular β -Sheet Forming Peptide Conjugated with Near-Infrared Chromophore for Selective Targeting, Imaging, and Dysfunction of Mitochondria. *Bioconjug Chem* **31**, 1301-1306 (2020).
- 14 Cheng DB, et al. Endogenous Reactive Oxygen Species-Triggered Morphology Transformation for Enhanced Cooperative Interaction with Mitochondria. *J Am Chem Soc.* **141**, 7235-7239 (2019).
- 15 Pederzoli F, et al. Nanomedicine Against A β Aggregation by β -Sheet Breaker Peptide Delivery: In Vitro Evidence. *Pharmaceutics* **11**, 572 (2019).
- 16 Luo S, et al. Targeting self-assembly peptide for inhibiting breast tumor progression and metastasis. *Biomaterials* **249**, 120055 (2020).
- 17 Versluis F, van Esch JH, Eelkema R. Synthetic Self-Assembled Materials in Biological Environments. *Adv Mater.* **28**, 4576-92 (2016).
- 18 Ben-Nun Y, et al. Cathepsin nanofiber substrates as potential agents for targeted drug delivery. *J Control Release* **257**, 60-67 (2017).
- 19 Mao L, et al. Conjugation of two complementary anti-cancer drugs confers molecular hydrogels as a co-delivery system. *Chem Commun (Camb)* **48**, 395-7 (2012).
- 20 Wang H, et al. Integrating Enzymatic Self-Assembly and Mitochondria Targeting for Selectively Killing Cancer Cells without Acquired Drug Resistance. *J Am Chem Soc* **138**, 16046-16055 (2016).
- 21 Gao Y, Shi J, Yuan D, Xu B. Imaging enzyme-triggered self-assembly of small molecules inside live cells. *Nat Commun.* **3**, 1033 (2012).
- 22 Brooks PC, et al. Localization of matrix metalloproteinase MMP-2 to the surface of invasive cells by interaction with integrin α v β 3. *Cell* **85**, 683-693 (1996).
- 23 Egeblad M, Werb Z. New functions for the matrix metalloproteinases in cancer progression. *Nat Rev Cancer* **2**, 161-74 (2002).
- 24 Costa-Silva B, et al. Pancreatic cancer exosomes initiate pre-metastatic niche formation in the liver. *Nat Cell Biol.* **17**, 816-26 (2015).
- 25 Houg DS, Bijlsma MF. The hepatic pre-metastatic niche in pancreatic ductal adenocarcinoma. *Mol Cancer* **17**, 95 (2018).
- 26 Zhao S, et al. Highly-metastatic colorectal cancer cell released miR-181a-5p-rich extracellular vesicles promote liver metastasis by activating hepatic stellate cells and remodelling the tumour microenvironment. *J Extracell Vesicles* **11**, e12186 (2022).
- 27 Kaplan RN, et al. VEGFR1-positive haematopoietic bone marrow progenitors initiate the pre-metastatic niche. *Nature* **438**, 820-827 (2005).

-
- ²⁸ Peinado H, et al. Pre-metastatic niches: organ-specific homes for metastases. *Nat Rev Cancer* **17**, 302-317 (2017).
- ²⁹ Aguado BA, Bushnell GG, Rao SS, Jeruss JS, Shea LD. Engineering the pre-metastatic niche. *Nat Biomed Eng* **1**, 0077 (2017).
- ³⁰ Erler, JT, et al. Hypoxia-induced lysyl oxidase is a critical mediator of bone marrow cell recruitment to form the pre-metastatic niche. *Cancer Cell* **6**, 35–44 (2009).
- ³¹ Peinado H, et al. Pre-metastatic niches: organ-specific homes for metastases. *Nat Rev Cancer* **17**, 302-317 (2017).
- ³² Zhang Z, Ai S, Yang Z, Li X. Peptide-based supramolecular hydrogels for local drug delivery. *Adv Drug Deliv Rev.* **174**, 482-503 (2021)..
- ³³ Zhao XX, et al. In Situ Self-Assembled Nanofibers Precisely Target Cancer-Associated Fibroblasts for Improved Tumor Imaging. *Angew Chem Int Ed Engl.* **58**, 15287-15294 (2019).
- ³⁴ Cascinelli N, Belli F, Mascheroni L, Lenisa L, Clemente C. Evaluation of clinical efficacy and tolerability of intravenous high dose thymopentin in advanced melanoma patients. *Melanoma Res.* **8**, 83-9 (1998).
- ³⁵ Zeng FL, et al. Clinical efficacy and safety of synthetic thymic peptides with chemotherapy for non-small cell lung cancer in China: A systematic review and meta-analysis of 27 randomized controlled trials following the PRISMA guidelines. *Int Immunopharmacol.* **75**, 105747 (2019).
- ³⁶ Kaplan RN, et al. VEGFR1-positive haematopoietic bone marrow progenitors initiate the pre-metastatic niche. *Nature* **438**, 820-827 (2005).
- ³⁷ Gabrilovich DI, et al. The terminology issue for myeloid-derived suppressor cells. *Cancer Res.* **67**, 425 author reply 6 (2007).
- ³⁸ Gabrilovich, DI. Myeloid-derived suppressor cells. *Cancer Immunol. Res.* **5**, 3-8 (2017).
- ³⁹ Erler, J. T. et al. Hypoxia-induced lysyl oxidase is a critical mediator of bone marrow cell recruitment to form the premetastatic niche. *Cancer Cell* **15**, 35-44 (2009).
- ⁴⁰ Wang, Z. et al. Periostin promotes immunosuppressive premetastatic niche formation to facilitate breast tumour metastasis. *J. Pathol.* **239**, 484-495 (2016).
- ⁴¹ Long, Y. et al. Self-delivery micellar nanoparticles prevent premetastatic niche formation by interfering with the early recruitment and vascular destruction of granulocytic myeloid-derived suppressor cells. *Nano Lett.* **20**, 2219-2229 (2020).
- ⁴² Jiang, T. et al. Metformin and Docosahexaenoic Acid Hybrid Micelles for Premetastatic Niche Modulation and Tumor Metastasis Suppression. *Nano lett.* **19**, 3548-3562 (2019).
- ⁴³ Lu, Z et al. Epigenetic therapy inhibits metastases by disrupting premetastatic niches. *Nature* **579**, 284-290 (2020).
- ⁴⁴ Kaczanowska, S. et al. Genetically engineered myeloid cells rebalance the core immune suppression program in metastasis. *Cell* **184**, 2033-2052 (2021).
- ⁴⁵ Kong J, et al. Extracellular vesicles of carcinoma-associated fibroblasts creates a pre-metastatic niche in the lung through activating fibroblasts. *Molecular Cancer* **18**, 175 (2019).

-
- ⁴⁶ Pein M, et al. Metastasis-initiating cells induce and exploit a fibroblast niche to fuel malignant colonization of the lungs. *Nat Commun.* **11**, 1494 (2020).
- ⁴⁷ Zhou X, et al. Melanoma cell-secreted exosomal miR-155-5p induce proangiogenic switch of cancer-associated fibroblasts via SOCS1/JAK2/STAT3 signaling pathway. *J. Exp. Clin. Cancer Res.* **37**, 242 (2018).
- ⁴⁸ Gabrilovich DI, et al. The terminology issue for myeloid-derived suppressor cells. *Cancer Res.* **67**, 425 author reply 6 (2007).
- ⁴⁹ Gabrilovich, DI. Myeloid-derived suppressor cells. *Cancer Immunol. Res.* **5**, 3-8 (2017).
- ⁵⁰ Lardner A. The effects of extracellular pH on immune function. *J Leukoc Biol.* **69**, 522-30 (2001).
- ⁵¹ Riemann A, et al. Acidic environment activates inflammatory programs in fibroblasts via a cAMP-MAPK pathway. *Biochim Biophys Acta* **1853**, 299-307 (2015).
- ⁵² Xiong ZG, et al. Neuroprotection in ischemia: blocking calcium-permeable acid-sensing ion channels. *Cell* **118**, 687-98 (2004).
- ⁵³ Gatenby RA, Gillies RJ. Why do cancers have high aerobic glycolysis? *Nat Rev Cancer.* **4**, 891-9 (2004).
- ⁵⁴ Kovalainen M, et al. Novel delivery systems for improving the clinical use of peptides. *Pharmacol Rev.* **67**, 541-61 (2015).
- ⁵⁵ Diao L, Meibohm B. Pharmacokinetics and pharmacokinetic-pharmacodynamic correlations of therapeutic peptides. *Clin Pharmacokinet.* **52**, 855-68 (2013).
- ⁵⁶ Sennello LT, et al. Single-dose pharmacokinetics of leuprolide in humans following intravenous and subcutaneous administration. *J Pharm Sci.* **75**, 158-60 (1986).
- ⁵⁷ Wills RJ, Dennis S, Spiegel HE, Gibson DM, Nadler PI. Interferon kinetics and adverse reactions after intravenous, intramuscular, and subcutaneous injection. *Clin Pharmacol Ther.* **35**, 722-7 (1984).
- ⁵⁸ Petros WP. Pharmacokinetics and administration of colony-stimulating factors. *Pharmacotherapy* **12**, 32S-38S (1992).
- ⁵⁹ Mannucci PM, et al. Intravenous and subcutaneous administration of desmopressin (DDAVP) to hemophiliacs: pharmacokinetics and factor VIII responses. *Thromb Haemost.* **58**, 1037-9 (1987).

REVIEWERS' COMMENTS

Reviewer #1 (Remarks to the Author):

This manuscript has improved considerably, although the evidence of nano-blanket could be sounder. In principle, I would support the acceptance of this work. It, however, baffles me that the authors appear to be evasive on referencing the prior works done on FFKY. The authors wrote the following paragraph in the rebuttal to acknowledge the prior use of FFKY, but still refused to reference those works properly. I think this could be ethically problematic. I would suggest the authors properly reference the use of FFKY in the first place when FFKY appears in main text.

“In addition, as the original fragment of A β 16-20, KLVFF has been applied as therapeutic agents^{7, 12}, imaging agents¹³ or delivery platform¹⁴ in the field of Alzheimer's disease¹⁵ and cancer¹⁶. Further researches cut the peptide fragment down to dipeptide FF to obtain discrete nanotubes through self-assembly in aqueous solution¹. Afterwards, felicitous modification and re-designment on FF with perfect biocompatibility has been reported with a wide range of applications¹⁷. For example, FFKF was developed to construct drug delivery system which can be complete degraded by cathepsin proteases¹⁸. Yang and his team employed FFFK to develop molecular hydrogel for codelivery of anti-cancer drugs¹⁹. In another work, the application of FFYK was explored in organelles targeting and cancer cell killing²⁰. Some introduced naphthyl group on N terminal of Phe to favor intercellular hydrophobic interactions²¹. Here in our manuscript, taking advantages of both the self-assembly feature of diphenylalanine structural motif FF with the assistance of Y and the editable site provided by K to combine with hydrophilic fragment via enzyme-cleavable linker, FFKY was employed as the backbone of the self-assembled monomer FG8, which would spontaneously fold into lamellar structure, constructing the peptide nano-blanket.”

Reviewer #3 (Remarks to the Author):

The authors have overall answered all of my concerns.

This reviewer would think that Figure II. In the rebuttal letter as well as organ distribution in the lung, liver, and other organs should be placed in the paper.

Also, pseudo coloring helped the visualization of the nanoblanket for researchers outside of the field. It would be nice to include an image with pseudo coloring as well on the side.

** See Nature Portfolio's author and referees' website at www.nature.com/authors for information about policies, services and author benefits

Reviewer #1 (Remarks to the Author):

This manuscript has improved considerably, although the evidence of nano-blanket could be sounder. In principle, I would support the acceptance of this work. It, however, baffles me that the authors appear to be evasive on referencing the prior works done on FFKY. The authors wrote the following paragraph in the rebuttal to acknowledge the prior use of FFKY, but still refused to reference those works properly. I think this could be ethically problematic. I would suggest the authors properly reference the use of FFKY in the first place when FFKY appears in main text.

“In addition, as the original fragment of A β ₁₆₋₂₀, KLVFF has been applied as therapeutic agents^{7, 12}, imaging agents¹³ or delivery platform¹⁴ in the field of Alzheimer's disease¹⁵ and cancer¹⁶. Further researches cut the peptide fragment down to dipeptide FF to obtain discrete nanotubes through self-assembly in aqueous solution¹. Afterwards, felicitous modification and re-designment on FF with perfect biocompatibility has been reported with a wide range of applications¹⁷. For example, FFKF was developed to construct drug delivery system which can be completely degraded by cathepsin proteases¹⁸. Yang and his team employed FFFK to develop molecular hydrogel for codelivery of anti-cancer drugs¹⁹. In another work, the application of FFYK was explored in organelles targeting and cancer cell killing²⁰. Some introduced naphthyl group on N terminal of Phe to favor intercellular hydrophobic interactions²¹. Here in our manuscript, taking advantages of both the self-assembly feature of diphenylalanine structural motif FF with the assistance of Y and the editable site provided by K to combine with hydrophilic fragment via enzyme-cleavable linker, FFKY was employed as the backbone of the self-assembled monomer FG8, which would spontaneously fold into lamellar structure, constructing the peptide nano-blanket.”

Response: Thank you for your suggestion. To fully present the entire process of how the idea of FR17 was formed based on FFKY to our readership, we've introduced the origin and development of the backbone of the assembly peptide, FFKY, in the Introduction section as follows:

“One of the research branches, which has been widely applied, was based on the β -sheet regions of Amyloid- β (A β)³⁰. As the original fragment of A β ₁₆₋₂₀, KLVFF has been applied as therapeutic agents^{26, 31}, imaging agents³² or delivery platform³³ in the field of Alzheimer's disease³⁴ and cancer³⁵. Further researches cut the peptide fragment down to dipeptide FF to obtain discrete nanotubes through self-assembly in aqueous solution³⁶. Afterwards, felicitous modification and re-designment on FF with perfect biocompatibility has been reported with a wide range of applications³⁷. For example, FFKF was developed to construct drug delivery system which can be completely degraded by cathepsin proteases²⁷. Yang and his team employed FFFK to develop molecular hydrogel for codelivery of anti-cancer drugs³⁸. In another work, the application of FFYK was explored in organelles targeting and cancer cell killing³⁹. Some introduced naphthyl group on N terminal of Phe to favor intercellular hydrophobic interactions of FFKY⁴⁰.”

“The self-assembly feature of FG8 is provided by its backbone, FFKY, taking advantages of both

the self-assembly feature of diphenylalanine structural motif FF with the assistance of Y and the editable site provided by K to combine with hydrophilic fragment *via* enzyme-cleavable linker.”

Reviewer #3 (Remarks to the Author):

The authors have overall answered all of my concerns.

This reviewer would think that Figure II in the rebuttal letter as well as organ distribution in the lung, liver, and other organs should be placed in the paper.

Also, pseudo coloring helped the visualization of the nano-blanket for researchers outside of the field. It would be nice to include an image with pseudo coloring as well on the side.

Response: We appreciate for your acknowledgement on our work. According to your suggestion, Figure II in the last response letter and inserted in the paper as Supplementary Figure 5. The pseudo coloring images have been inserted into Figure 2 to give a direct impression of the morphology of peptide nano-blanket on the readership as follow.

Supplementary Figure 5. Enzyme-activated assembly of FR17 and sFD17. **a**, Cryo-TEM image of the peptide nano-blanket assembled by FR17 (500 μ M) treated with MMP2 (1 μ g/ml) for 24 h. Scale bar = 200 nm. **b**, Macroscopic images of the thin layer formed by FR17 treated with enzyme and let stand for 72 h. The soft thin layer broke into pieces after gently shaking and

dispersed into nanoscale fragments after sharply shaking. **c**, STEM images of the peptide nano-blanket in the intercellular substance in PMN lung, which was collected from PMN mouse at 12 h-post subcutaneous administration of FR17. Scale bar = 1 μm . The lamellar structure of the peptide nano-blanket was pseudo-colored in gold.

Figure 2. Enzyme-activated self-assembly of FR17 and all-atom molecular dynamics (MD) simulation of the self-assembly of FG8. **a**, TEM images of FR17 or sFD17 (500 μM) treated with MMP2 (scale bar = 200 nm), FG8 (500 μM , scale bar = 100 nm) and FFKY (500 μM , scale bar = 100 nm). The lamellar structure of the peptide nano-blanket was pseudo-colored in gold. **b**, Size distribution of FR17 and sFD17 (500 μM) after MMP2 cleavage. **c**, Molecular structure of FFKY. **d**, The FFKY assemblies generated at $t = 200$ ns of MD simulation of 16 FFKY molecules in water (containing NaCl for charge neutralization). **e**, Typical molecular cluster and the non-bonded interactions involved in FFKY assemblies in detail. **f**, Molecular structure of FG8. **g**, The FG8 assemblies generated at $t = 200$ ns of MD simulation of 16 FG8 molecules in water (containing NaCl for charge neutralization). **h**, Typical molecular cluster and the non-bonded interactions involved in FG8 assemblies in detail. The dashed purple lines denote the π - π stacking. The dashed blue lines denote the hydrogen bonds. The nitrogen atoms are labeled in blue. And the oxygen atoms are labeled in red. Source data are provided as a Source Data file.

In addition, organ distribution of peptide nano-blanket indicated by the aggregation-induced

emission (AIE) effect observed in different organs' sections at 12 h-post TPE-FR17 administration has been presented in Supplementary Fig. 7-8 as follow. Besides, the further discussion on the peptide assemblies observed both in the PMN lung and liver, as well as the time-dependent aggregation and degradation of peptide nano-blanket is achievable in the last response letter, which will be revealed to the readership accompanied the paper as Peer Review File.

“Fluorescent dots observed in the organs' sections revealed the responsive-assembly of the peptide nano-blanket in PMN lung *in vivo* (Supplementary Fig. 7) with few aggregations in PMN liver and no observation of AIE effect in other organs (Supplementary Fig. 8).”

Supplementary Figure 7. The aggregation-induced emission effect of the *in-situ* assembly of peptide nano-blanket in the lung *in vivo*. Lung sections at 12 h-post subcutaneous administration of TPE-FR17 (100 μ M/kg) in healthy or PMN lung. Peptide assemblies of the monomer TPE-FG8 released from TPE-FR17 in PMN were pseudo-colored in white on the upper panel and pseudo-colored in red in the merged images, and indicated by red arrows for contrast. Scale bar = 200 μ m.

Supplementary Figure 8. The aggregation-induced emission effect of the *in-situ* assembly of peptide nano-blanket in heart, spleen, kidney and liver *in vivo*. Organs were collected at 12 h-post subcutaneous administration of TPE-FR17 (100 μ M/kg) to healthy or PMN mice. Peptide assemblies of the monomer TPE-FG8 released from TPE-FR17 in PMN were pseudo-colored in white on the upper panel and pseudo-colored in red in the merged images, and indicated by red arrows for contrast. Scale bar = 200 μ m.